# TimeEmb: A Lightweight Static-Dynamic Disentanglement Framework for Time Series Forecasting

**Mingyuan Xia**[*]
Jilin University
xiamy2322@mails.jlu.edu.cn

**Chunxu Zhang**[*]
Jilin University
zhangchunxu@jlu.edu.cn

**Zijian Zhang**[†]
Jilin University
zhangzijian@jlu.edu.cn

**Hao Miao**
The Hong Kong Polytechnic University
hao.miao@polyu.edu.hk

**Qidong Liu**
Xi'an Jiaotong University
liuqidong@xjtu.edu.cn

**Yuanshao Zhu**
City University of Hong Kong
yuanshao@ieee.org

**Bo Yang**
Jilin University
ybo@jlu.edu.cn

## Abstract

Temporal non-stationarity, the phenomenon that time series distributions change over time, poses fundamental challenges to reliable time series forecasting. Intuitively, the complex time series can be decomposed into two factors, *i.e.,* time-invariant and time-varying components, which indicate static and dynamic patterns, respectively. Nonetheless, existing methods often conflate the time-varying and time-invariant components, and jointly learn the combined long-term patterns and short-term fluctuations, leading to suboptimal performance facing distribution shifts. To address this issue, we initiatively propose a lightweight static-dynamic decomposition framework, TimeEmb, for time series forecasting. TimeEmb innovatively separates time series into two complementary components: (1) time-invariant component, captured by a novel global embedding module that learns persistent representations across time series, and (2) time-varying component, processed by an efficient frequency-domain filtering mechanism inspired by full-spectrum analysis in signal processing. Experiments on real-world datasets demonstrate that TimeEmb outperforms state-of-the-art baselines and requires fewer computational resources. We conduct comprehensive quantitative and qualitative analyses to verify the efficacy of static-dynamic disentanglement. This lightweight framework can also improve existing time-series forecasting methods with simple integration. To ease reproducibility, the code is available at https://github.com/showmeon/TimeEmb.

## 1 Introduction

The proliferation of edge devices and mobile sensing results in a large amount of time series data, enabling various real-world applications [58, 51, 15, 26]. In this study, we focus on time series forecasting, which plays a pivotal role in decision-making across critical domains including energy management [11], transportation systems [57, 7], and financial markets [44].

---

[*]These authors contributed equally to this work.

[†]Corresponding author.

39th Conference on Neural Information Processing Systems (NeurIPS 2025).

Traditional statistical methods, *e.g.,* ARMA [2], employ moving average techniques to model temporal dependencies. With the advances of neural networks, deep learning methods have revolutionized temporal pattern extraction, delivering superior performance. These methods include recurrent neural networks (RNNs) models [13, 35] that capture sequential dynamics, convolutional neural networks (CNNs) models [8, 18] to obtain hierarchical features, and transformer-based models [46] to learn long-range dependencies with self-attention mechanisms. Recently, Multi-layer perceptron (MLP) methods [56] have demonstrated their effectiveness and superior efficiency compared to transformer-based counterparts.

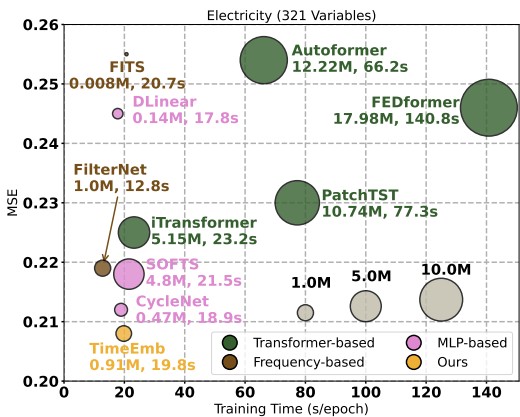

Figure 1: Efficiency and performance comparison on the Electricity dataset.

Despite the advancements of existing methods, a fundamental challenge persists in modeling complex temporal dependencies, *i.e.,* non-stationarity. Real-world time series often exhibit dynamic distribution shifts due to evolving trends and external interventions, showing high non-stationarity [32, 20]. This dynamic distribution shift violates the independent and identical distribution (IID) assumption of most existing forecasting methods [9]. It poses great challenges to robust temporal dependency modeling [38] and calls for a novel method that can handle such non-stationarity and learn comprehensive temporal dependencies.

Intuitively, time series can be considered as a combination of two complementary parts: static time-invariant and dynamic time-varying components [21, 36, 6]. **Time-invariant component** represents stable long-term patterns in time series. For example, traffic flow typically follows a regular pattern, with peaks in the morning and troughs at night. **Time-varying component** reflects local fluctuations in time series, *e.g.,* abnormal traffic flow caused by extreme weather or accidents. We contend that effective disentanglement of these two components can prevent the model from mistaking short-term noise for long-term patterns. It can explicitly capture stable long-term and dynamic local dependencies, thereby improving the effectiveness of time series forecasting.

However, it is non-trivial to develop this kind of model. In general, there remain three major limitations unsolved for time series disentanglement: **(1) Ignorance of long-term invariant pattern modeling.** Existing seasonal-trend disentanglement methods often generate the trend component by moving the average kernel, and consider the rest as the seasonal component [51, 56, 59, 25]. It is performed on local time series by smoothing [37], and can hardly learn the global static patterns in the whole time series. **(2) Rigorous assumption.** To pursue explicit disentanglement, some methods rely on strong assumptions that may not always hold in practice. For example, CycleNet [14] assumes a fixed periodic pattern in the dataset and extracts it using a learnable recurrent cycle. However, this assumption does not always hold, as the complex periodicities can vary or have diverse lengths. Moreover, relying on a pre-defined cycle length leads to limited flexibility and unstable efficacy. It cannot learn periodicity without providing an exact cycle length. **(3) High model complexity.** The quadratic complexity of the self-attention mechanism hinders practical application [51, 59]. As shown in Figure 1, Transformer-based methods exhibit relatively large model sizes and high training costs. Recent methods based on frequency analysis and MLP partially alleviate this with more efficient architectures. However, a satisfactory balance between performance and efficiency remains elusive.

To address these problems, we propose TimeEmb, a lightweight static-dynamic disentanglement framework. TimeEmb decomposes the original time series into time-invariant and time-varying components, and processes them accordingly. Specifically, we introduce a learnable time-invariant embedding bank to extract static time-invariant patterns. These embeddings are consistent across all time series segments within the entire dataset, aiming to capture long-term and stable temporal patterns. In addition, the embedding bank provides specific embedding for individual timesteps. This enables the model to adapt to local data distribution shifts since time-invariant patterns may differ at different timesteps. By separating the time-invariant component from the time series, we obtain the remaining time-varying component, illustrating dynamic disturbance. Frequency analysis describes complex signals using their intensity in the frequency spectrum [1], which presents clear intrinsic periodicity features. Inspired by this, we design an efficient frequency filter to process the

time-varying component through dense weighting. Based on the explicit decomposition and parallel processing of the static and dynamic components, TimeEmb achieves state-of-the-art performance. Meanwhile, due to its lightweight architecture, it requires fewer computational resources. As shown by its optimal position in Figure 1, the proposed TimeEmb strikes an excellent balance between performance and efficiency.

Our major contributions are summarized as follows:

- For the first time, we propose to leverage a learnable embedding bank to capture the global recurrent features while adapting to local distribution shifts.

- We propose TimeEmb, which explicitly disentangles the time series and systematically addresses the time-invariant component using a learnable embedding bank and time-varying component via frequency filtering.

- The proposed TimeEmb can easily and seamlessly serve as a plug-in to enhance existing methods with minimum additional computational cost.

- Experiments on seven benchmark datasets from diverse scenarios demonstrate the superior performance of the proposed TimeEmb. TimeEmb is efficient in terms of computation and storage compared to existing state-of-the-art baselines.

## 2   Related Work

***Transformer-based Time Series Forecasting.*** Transformers have shown strong sequence modeling capabilities in time series forecasting [51, 58, 22]. PatchTST [28] segments sequences into fixed-length patches for local-global modeling, while iTransformer [22] and Informer [58] reduce attention complexity to improve scalability. However, attention-based models still incur considerable computational and memory costs [12, 43], limiting deployment in resource-constrained settings. In contrast, TimeEmb leverages lightweight spectral modules, *i.e.,* including an embedding bank and frequency filter, to achieve strong performance with reduced overhead.

***MLP-based Time Series Forecasting.*** Recently, MLP methods, *e.g.,* TSMixer [4] and TimeMixer [49], have demonstrated competitive forecasting performance with reduced complexity. DLinear [56] further improves efficiency by separating trend and residual components. By contrast, TimeEmb provides an explicit disentanglement framework in the frequency domain, enabling simultaneous modeling of time-invariant and time-varying patterns beyond what time-domain MLPs can express.

***Frequency-based Time Series Forecasting.*** Recent work has explored Fourier-based representations to model periodicity and reduce noise sensitivity [52, 59, 29]. While most methods apply global spectral analysis, TimeEmb introduces a fine-grained disentanglement strategy: a time-invariant component is learned across the full spectrum via embedding, while the dynamic part is filtered adaptively by a learnable frequency modulation. This structured spectral design extends the utility of frequency-domain modeling for complex time series.

***Embedding-enhanced Forecasting.*** Embedding strategies have been adopted to encode positional, spatial, or temporal context [28, 39, 10]. For instance, STID [39] and D2STGNN [41] use spatiotemporal embeddings, while SOFTS [10] shares embeddings across channels. Unlike these, TimeEmb establishes a learnable temporal embedding bank that captures global time-invariant patterns across the dataset, with each embedding specializing in a specific time slot to model static structures in a data-driven and frequency-aware manner.

***LLM-based Time Series Forecasting.*** With the rapid development of Large Language Models, recent research has explored their potential in time series forecasting by treating temporal signals as sequential tokens [19, 16]. LLM-based approaches benefit from strong generalization and transfer capabilities, enabling zero-shot or few-shot forecasting across diverse domains [17, 24, 48]. Nevertheless, these models are typically resource-intensive and require massive pretraining corpora, making them impractical for lightweight or domain-specific applications. Compared with these paradigms, TimeEmb focuses on a compact yet effective disentanglement mechanism that achieves comparable forecasting accuracy with substantially reduced computational cost and training complexity.

# 3 Methodology

## 3.1 Framework Overview

Given historical time series $\boldsymbol{X} \in \mathbb{R}^{L \times D}$ with $L$ timesteps and $D$ channels, time series prediction aims to infer the future states of $H$ timesteps, *i.e.*, $\widehat{\boldsymbol{X}} \in \mathbb{R}^{H \times D}$.

TimeEmb addresses time series forecasting via disentangled representation learning in the frequency domain. The core idea is to decompose the input sequence into a time-invariant component $\boldsymbol{X}_s$ and a time-varying component $\boldsymbol{X}_d$.

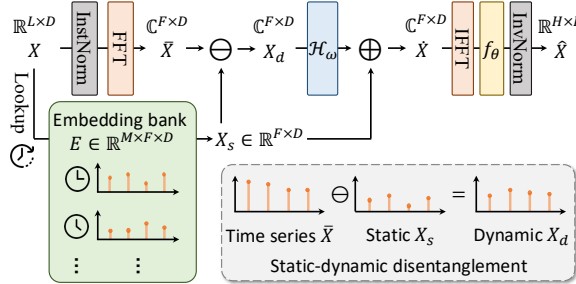

Figure 2: TimeEmb framework overview.

Specifically, we first transform the input series $\boldsymbol{X}$ into its frequency representation $\overline{\boldsymbol{X}}$ via the Fourier transform. Then, we retrieve $\boldsymbol{X}_s$ from a learnable embedding bank $\boldsymbol{E}$ based on the input timestamp, capturing long-term stable patterns. The dynamic part $\boldsymbol{X}_d$ is obtained by subtracting $\boldsymbol{X}_s$ from $\overline{\boldsymbol{X}}$. To model complex dynamics, we apply a learnable frequency filter $\mathcal{H}_\omega$ to $\boldsymbol{X}_d$, emphasizing informative frequencies and suppressing noise. The filtered dynamic and static components are then fused and transformed back to the time domain for final prediction. This frequency-based decomposition allows TimeEmb to efficiently capture both periodic structures and transient variations in a lightweight and interpretable manner.

## 3.2 Domain Transformation

Viewing time series data from the perspective of the frequency domain offers unique insights into its underlying structure. Unlike the time domain, where patterns may be obscured by noise or nonlinearity, the frequency spectrum reveals the distribution of different periodic components and their relative energy contributions. Transforming time series into the frequency domain decomposes it into distinct frequency components, describing the complex signal as a linear combination of sine and cosine waves with varying frequencies and amplitudes. This process helps reveal underlying periodicities and hidden features that are otherwise obscure in the time domain [42].

Given a discrete temporal sequence $\boldsymbol{X} \in \mathbb{R}^{L \times D}$ (we consider the univariate case with $D = 1$ for clarity), we first conduct instance normalization InstNorm() to standardize each instance's distribution at every timestep. Then, its frequency-domain representation $\overline{\boldsymbol{X}} \in \mathbb{C}^{F \times D}$ can be obtained using real-valued Fast Fourier Transform (rFFT) [27],

$$\overline{\boldsymbol{X}}[k] = \sum_{n=0}^{L-1} \boldsymbol{X}[n] e^{-j2\pi kn/L}, \quad k = 0, 1, ..., F-1, \tag{1}$$

where $j = \sqrt{-1}$ is the imaginary unit. Due to the conjugate symmetry property of real signals in the Fourier domain, the number of unique frequency components is $F = \lfloor L/2 \rfloor + 1$, allowing for a compact representation without redundancy.

## 3.3 Static Component via Embedding Bank

Existing approaches to modeling time-invariant patterns, such as seasonal-trend decomposition [51, 56], typically divide a time series into trend and residual components using local smoothing methods. However, this method merely considers locally stable and dynamic parts in the input time series, and fails to uncover the long-standing invariant features in the dataset. Recently, CycleNet [14] attempts to address this by learning a periodic embedding, but it depends on a predefined period length from expert knowledge, and slight changes can cause severe performance drops.

To address these limitations, TimeEmb proposes a flexible and learnable mechanism to capture long-term, recurrent patterns shared across time series through a temporal embedding bank. For example, in traffic forecasting, we aim to capture recurring daily structures such as typical rush-hour

patterns. Since intra-day patterns also vary over time (*e.g.,* hourly traffic flow fluctuations), we construct embeddings for each timestep.

In specific, we define a learnable embedding bank $\boldsymbol{E} \in \mathbb{R}^{M \times F \times D}$ consisting of $M$ embeddings to preserve invariant patterns in a day. $M$ controls the granularity of intra-day specific patterns. For instance, when $M = 24$, $\boldsymbol{E}$ assigns an embedding to each hour; when $M = 96$, it captures common patterns every 15 minutes. To guarantee the embedding learns the general pattern across the time series, we leverage the last timestep of the input $\boldsymbol{X}$ as $t_{last}$. This index enables us to retrieve embedding from $\boldsymbol{E}$, *i.e.,* $\boldsymbol{X}_s = \boldsymbol{E}[t_{last} \bmod M]$. Then, we separate the embedding $\boldsymbol{X}_s$ from time series $\overline{\boldsymbol{X}}$ and obtain the time-varying component $\boldsymbol{X}_d$ as follows,

$$\boldsymbol{X}_d = \overline{\boldsymbol{X}} - \boldsymbol{X}_s. \tag{2}$$

In this operation, we subtract the real number $\boldsymbol{X}_s$ from the real part of the complex number $\overline{\boldsymbol{X}}$, which can reduce the computational and storage cost of the embedding bank. The embedding bank $\boldsymbol{E}$ is optimized across the entire dataset and learns to encode consistent patterns that emerge at the same time across different days. For instance, when $M = 24$, each embedding is tuned to capture the average behavior at a specific hour of the day (*e.g.,* peaks around 8:00 and lows around 23:00), enabling the model to represent both the global temporal structure and local variations. Importantly, this embedding structure is flexible: while we focus on day-level periodicities, it can be naturally extended to model weekly or other common-sense periods by adjusting $M$, without relying on domain knowledge. This design enables TimeEmb to learn shared expressive representations of time-invariant components, which are essential for disentangled modeling and robust generalization.

## 3.4 Dynamic Component via Frequency Filtering

To effectively model the dynamic component $\boldsymbol{X}_d$, we apply a learnable spectral filter in the frequency domain. This design is motivated by the Convolution Theorem [23], *i.e.,* circular convolution in the time domain is equivalent to element-wise multiplication in the frequency domain. Thus, frequency-domain filtering provides an efficient and expressive way to implement time-invariant linear operations on temporal signals.

We introduce a complex-valued spectral modulation vector $\boldsymbol{\omega} \in \mathbb{C}^{F \times 1}$, shared across channels, to selectively reweight different frequency bands. The filtering operation is defined as:

$$\mathcal{H}_{\boldsymbol{\omega}}(\boldsymbol{X}_d)[k] = \boldsymbol{X}_d[k] \odot \boldsymbol{\omega}[k], \tag{3}$$

where $\odot$ represents dot product.

This operation can be interpreted as learning the frequency response function of a linear time-invariant (LTI) system [50]. By optimizing $\omega$ end-to-end, the model can flexibly approximate linear time-invariant transformations of temporal signals. This provides both theoretical generality and practical flexibility for modeling diverse temporal dynamics. Theoretical analysis can be referred to **Appendix A**. After modulation, the filtered dynamic component is fused with the time-invariant part $\boldsymbol{X}_s$ to recover the full frequency representation:

$$\dot{\boldsymbol{X}} = \mathcal{H}_{\boldsymbol{\omega}}(\boldsymbol{X}_d) + \boldsymbol{X}_s. \tag{4}$$

## 3.5 Prediction Layer

We leverage a prediction layer $f_{\boldsymbol{\theta}}$ to produce the final prediction given representation $\dot{\boldsymbol{X}}$. It can be customized to specific requirements. We adopt a two-layer MLP architecture in TimeEmb,

$$f_{\boldsymbol{\theta}}(\boldsymbol{X}) = \boldsymbol{W}_2(\text{ReLU}(\boldsymbol{W}_1 \boldsymbol{X} + \boldsymbol{b}_1)) + \boldsymbol{b}_2, \tag{5}$$

where $\boldsymbol{W}_1 \in \mathbb{R}^{d \times L}$ and $\boldsymbol{W}_2 \in \mathbb{R}^{H \times d}$ are projection matrices, $H$ denotes the forecasting horizon, and $\boldsymbol{b}_1 \in \mathbb{R}^d$, $\boldsymbol{b}_2 \in \mathbb{R}^H$ are the corresponding biases.

To restore the time series to its original scale, we conduct inverse normalization with the instance-specific mean and variance. Consequently, the final prediction $\widehat{\boldsymbol{X}} \in \mathbb{R}^{H \times D}$ is computed as,

$$\widehat{X} = \text{InvNorm}(f_{\theta}(\text{IFFT}(\dot{X}))). \tag{6}$$

## 3.6 Optimization Objective

For model optimization, we employ the Mean Squared Error (MSE) to measure the loss between prediction and ground truth. Inspired by the self-correlation of the values in time series [47], we introduce Mean Absolute Error (MAE) loss in the frequency domain to alleviate the influence of self-correlation. In summary, our optimization objective function $\mathcal{L}$ can be expressed as follows,

$$\mathcal{L}(\widehat{X}, Y) = \alpha\text{MAE}(\text{FFT}(\widehat{X}), \text{FFT}(Y)) + (1 - \alpha)\text{MSE}(\widehat{X}, Y), \tag{7}$$

where $\alpha \in [0, 1]$ is hyper-parameter. The workflow of TimeEmb is detailed in **Appendix B**.

## 3.7 Computational Efficiency Analysis

We analyze the computational complexity of the core components of our TimeEmb, *i.e.,* time-invariant embedding bank and frequency filtering.

***Time-invariant Embedding.*** The embedding bank $E \in \mathbb{R}^{M \times F \times D}$ supports two lightweight operations: *embedding lookup* and *frequency-wise subtraction*. Given an input time series $X \in \mathbb{R}^{L \times D}$, an embedding $X_s \in \mathbb{R}^{F \times D}$ is retrieved based on its last timestamp with complexity $\mathcal{O}(M)$. The subtraction step $X_d = \overline{X} - X_s$ involves $\mathcal{O}(F \times D)$ operations. Thus, the overall complexity is linear, *i.e.,* $\mathcal{O}(M + F \times D)$.

The embedding bank is also parameter-efficient: for example, in ETTh1 with $L = 96$, $M = 24$, $F = 49$, and $D = 7$, the total parameters required are only $24 \times 49 \times 7 = 8,232$.

***Frequency Filtering.*** The spectral modulation of the dynamic component $X_d$ is performed by element-wise multiplication with the learnable filter $\omega \in \mathbb{C}^{F \times 1}$, yielding a complexity of $\mathcal{O}(F \times D)$.

Finally, the dominant cost in TimeEmb arises from the Fourier Transform, which operates at $\mathcal{O}(D \times L \log L)$. Overall, the computational complexity of key components is linear, making it highly efficient and scalable for long sequences and multivariate inputs.

## 4 Experiments

In this section, we conduct extensive experiments with real-world time series benchmarks to sufficiently assess the performance of our proposed model, including comparison with SOTA baselines (Section 4.2), compatibility evaluation (Section 4.3), time series disentanglement capability analysis (Section 4.4), and modules' effectiveness verification (Section 4.5).

### 4.1 Experimental Setup

#### 4.1.1 Datasets and Baselines

Following the mainstream evaluation setup in existing time series prediction studies [51, 58], we conduct experiments on seven real-world benchmark datasets, including four ETT datasets (ETTh1, ETTh2, ETTm1, ETTm2) [58], Weather [51], Electricity (ECL) [51], and Traffic [51]. Following prior works [51, 22], we split the ETTs dataset into training, validation, and test sets with a ratio of 6:2:2, while the other datasets were split in a ratio of 7:1:2.

To comprehensively evaluate the effectiveness, we select comprehensive SOTA baselines across three representative frameworks: (1) **Frequency-based models**: FilterNet [54], FITS [53], and FreTS [55]; (2) **MLP-based models**: DLinear [56] and CycleNet [14]; and (3) **Transformer-based models**: iTransformer [22], PatchTST [28], and Fredformer [34]. Detailed introduction of datasets and baselines can be found in **Appendix C**.

#### 4.1.2 Implementation Details

To ensure fair comparison, we adapt common experimental settings: lookback window lengths $L \in \{96, 336, 720\}$ and prediction lengths $H \in \{96, 192, 336, 720\}$ for all baselines across datasets [51,

Table 1: Performance comparison with prediction lengths $H \in \{96, 192, 336, 720\}$ and lookback window length $L = 96$. The best results are highlighted in **bold** and the second best are underlined.

| Model | Metric | TimeEmb (ours) MSE | TimeEmb (ours) MAE | CycleNet 2024 MSE | CycleNet 2024 MAE | Fredformer 2024 MSE | Fredformer 2024 MAE | FilterNet 2024 MSE | FilterNet 2024 MAE | iTransformer 2024 MSE | iTransformer 2024 MAE | PatchTST 2023 MSE | PatchTST 2023 MAE | FITS 2024 MSE | FITS 2024 MAE | FreTS 2023 MSE | FreTS 2023 MAE | DLinear 2023 MSE | DLinear 2023 MAE |
|---|---|---|---|---|---|---|---|---|---|---|---|---|---|---|---|---|---|---|---|
| ETTh1 96 | | **0.366**±0.001 | **0.387**±0.001 | 0.378 | 0.391 | 0.373 | 0.392 | 0.375 | 0.394 | 0.386 | 0.405 | 0.394 | 0.406 | 0.386 | 0.396 | 0.395 | 0.407 | 0.386 | 0.400 |
| ETTh1 192 | | **0.417**±0.001 | **0.416**±0.001 | 0.426 | 0.419 | 0.433 | 0.420 | 0.436 | 0.422 | 0.441 | 0.436 | 0.440 | 0.435 | 0.436 | 0.423 | 0.448 | 0.440 | 0.437 | 0.432 |
| ETTh1 336 | | **0.457**±0.001 | **0.436**±0.001 | 0.464 | 0.439 | 0.470 | 0.437 | 0.487 | 0.458 | 0.487 | 0.458 | 0.491 | 0.462 | 0.478 | 0.444 | 0.499 | 0.472 | 0.481 | 0.459 |
| ETTh1 720 | | **0.459**±0.002 | 0.460±0.001 | 0.461 | 0.460 | 0.467 | 0.456 | 0.474 | 0.469 | 0.503 | 0.491 | 0.487 | 0.479 | 0.502 | 0.495 | 0.558 | 0.532 | 0.519 | 0.516 |
| ETTh1 avg | | **0.425**±0.001 | **0.425**±0.001 | 0.432 | 0.427 | 0.435 | 0.426 | 0.440 | 0.432 | 0.454 | 0.447 | 0.453 | 0.446 | 0.451 | 0.440 | 0.475 | 0.463 | 0.456 | 0.452 |
| ETTh2 96 | | **0.277**±0.001 | **0.328**±0.001 | 0.285 | 0.335 | 0.293 | 0.342 | 0.292 | 0.343 | 0.297 | 0.349 | 0.288 | 0.340 | 0.295 | 0.350 | 0.309 | 0.364 | 0.333 | 0.387 |
| ETTh2 192 | | **0.356**±0.001 | **0.379**±0.001 | 0.373 | 0.391 | 0.371 | 0.389 | 0.369 | 0.395 | 0.380 | 0.400 | 0.376 | 0.395 | 0.381 | 0.396 | 0.395 | 0.425 | 0.477 | 0.476 |
| ETTh2 336 | | **0.400**±0.002 | 0.417±0.001 | 0.421 | 0.433 | 0.382 | 0.409 | 0.420 | 0.432 | 0.428 | 0.432 | 0.440 | 0.451 | 0.426 | 0.438 | 0.462 | 0.467 | 0.594 | 0.541 |
| ETTh2 720 | | **0.416**±0.001 | 0.437±0.002 | 0.453 | 0.458 | 0.415 | 0.434 | 0.430 | 0.446 | 0.427 | 0.445 | 0.436 | 0.453 | 0.431 | 0.446 | 0.721 | 0.604 | 0.831 | 0.657 |
| ETTh2 avg | | **0.362**±0.001 | **0.390**±0.001 | 0.383 | 0.404 | 0.365 | 0.393 | 0.378 | 0.404 | 0.383 | 0.407 | 0.385 | 0.410 | 0.383 | 0.408 | 0.472 | 0.465 | 0.559 | 0.515 |
| ETTm1 96 | | **0.304**±0.001 | **0.343**±0.001 | 0.319 | 0.360 | 0.326 | 0.361 | 0.318 | 0.358 | 0.334 | 0.368 | 0.329 | 0.365 | 0.355 | 0.375 | 0.335 | 0.372 | 0.345 | 0.372 |
| ETTm1 192 | | **0.354**±0.001 | **0.373**±0.001 | 0.360 | 0.381 | 0.363 | 0.383 | 0.364 | 0.383 | 0.377 | 0.391 | 0.377 | 0.394 | 0.392 | 0.393 | 0.380 | 0.389 | 0.380 | 0.389 |
| ETTm1 336 | | **0.379**±0.001 | **0.393**±0.001 | 0.389 | 0.403 | 0.395 | 0.403 | 0.396 | 0.406 | 0.426 | 0.420 | 0.400 | 0.410 | 0.424 | 0.414 | 0.421 | 0.426 | 0.413 | 0.413 |
| ETTm1 720 | | **0.435**±0.001 | **0.428**±0.001 | 0.447 | 0.441 | 0.453 | 0.438 | 0.456 | 0.444 | 0.491 | 0.459 | 0.475 | 0.453 | 0.487 | 0.449 | 0.486 | 0.465 | 0.474 | 0.453 |
| ETTm1 avg | | **0.368**±0.001 | **0.384**±0.001 | 0.379 | 0.396 | 0.384 | 0.395 | 0.384 | 0.398 | 0.407 | 0.410 | 0.396 | 0.406 | 0.415 | 0.408 | 0.408 | 0.416 | 0.403 | 0.407 |
| ETTm2 96 | | **0.163**±0.001 | **0.242**±0.001 | 0.163 | 0.246 | 0.177 | 0.259 | 0.174 | 0.257 | 0.180 | 0.264 | 0.184 | 0.264 | 0.183 | 0.266 | 0.189 | 0.277 | 0.193 | 0.292 |
| ETTm2 192 | | **0.226**±0.001 | **0.285**±0.001 | 0.229 | 0.290 | 0.243 | 0.301 | 0.240 | 0.300 | 0.250 | 0.309 | 0.246 | 0.306 | 0.247 | 0.305 | 0.258 | 0.326 | 0.284 | 0.362 |
| ETTm2 336 | | 0.286±0.001 | **0.324**±0.001 | 0.284 | 0.327 | 0.302 | 0.340 | 0.297 | 0.339 | 0.311 | 0.348 | 0.308 | 0.346 | 0.307 | 0.342 | 0.343 | 0.390 | 0.369 | 0.427 |
| ETTm2 720 | | **0.383**±0.001 | **0.381**±0.001 | 0.389 | 0.391 | 0.397 | 0.396 | 0.392 | 0.393 | 0.412 | 0.407 | 0.409 | 0.402 | 0.407 | 0.399 | 0.495 | 0.480 | 0.554 | 0.522 |
| ETTm2 avg | | **0.265**±0.001 | **0.308**±0.001 | 0.266 | 0.314 | 0.279 | 0.324 | 0.276 | 0.322 | 0.288 | 0.332 | 0.287 | 0.330 | 0.286 | 0.328 | 0.321 | 0.368 | 0.350 | 0.401 |
| Weather 96 | | **0.150**±0.001 | **0.190**±0.001 | 0.158 | 0.203 | 0.163 | 0.207 | 0.162 | 0.207 | 0.174 | 0.214 | 0.176 | 0.217 | 0.166 | 0.213 | 0.174 | 0.250 | 0.196 | 0.255 |
| Weather 192 | | **0.200**±0.001 | **0.238**±0.001 | 0.207 | 0.247 | 0.211 | 0.251 | 0.210 | 0.250 | 0.221 | 0.254 | 0.221 | 0.256 | 0.213 | 0.254 | 0.219 | 0.250 | 0.237 | 0.296 |
| Weather 336 | | **0.259**±0.001 | **0.282**±0.001 | 0.262 | 0.289 | 0.267 | 0.292 | 0.265 | 0.290 | 0.278 | 0.296 | 0.275 | 0.296 | 0.269 | 0.294 | 0.273 | 0.290 | 0.283 | 0.335 |
| Weather 720 | | **0.339**±0.001 | **0.336**±0.001 | 0.344 | 0.344 | 0.343 | 0.341 | 0.342 | 0.340 | 0.358 | 0.347 | 0.352 | 0.346 | 0.346 | 0.343 | 0.334 | 0.332 | 0.345 | 0.381 |
| Weather avg | | **0.237**±0.001 | **0.262**±0.001 | 0.243 | 0.271 | 0.246 | 0.272 | 0.245 | 0.272 | 0.258 | 0.278 | 0.256 | 0.279 | 0.249 | 0.276 | 0.250 | 0.270 | 0.265 | 0.317 |
| Electricity 96 | | **0.136**±0.001 | 0.231±0.001 | 0.136 | 0.229 | 0.147 | 0.241 | 0.147 | 0.245 | 0.148 | 0.240 | 0.164 | 0.251 | 0.200 | 0.278 | 0.176 | 0.258 | 0.197 | 0.282 |
| Electricity 192 | | 0.153±0.001 | 0.246±0.001 | 0.152 | 0.244 | 0.165 | 0.258 | 0.160 | 0.250 | 0.162 | 0.253 | 0.173 | 0.262 | 0.200 | 0.280 | 0.175 | 0.262 | 0.196 | 0.285 |
| Electricity 336 | | 0.170±0.001 | **0.264**±0.001 | 0.170 | 0.264 | 0.177 | 0.273 | 0.173 | 0.267 | 0.178 | 0.269 | 0.190 | 0.279 | 0.214 | 0.295 | 0.185 | 0.278 | 0.209 | 0.301 |
| Electricity 720 | | **0.208**±0.001 | **0.297**±0.001 | 0.212 | 0.299 | 0.213 | 0.304 | 0.210 | 0.309 | 0.225 | 0.317 | 0.230 | 0.313 | 0.255 | 0.327 | 0.220 | 0.315 | 0.245 | 0.333 |
| Electricity avg | | **0.167**±0.001 | 0.260±0.001 | 0.168 | **0.259** | 0.175 | 0.269 | 0.173 | 0.268 | 0.178 | 0.270 | 0.189 | 0.276 | 0.217 | 0.295 | 0.189 | 0.278 | 0.212 | 0.300 |
| Traffic 96 | | 0.432±0.002 | 0.279±0.001 | 0.458 | 0.296 | 0.406 | 0.277 | 0.430 | 0.294 | **0.395** | **0.268** | 0.427 | 0.272 | 0.651 | 0.391 | 0.593 | 0.378 | 0.650 | 0.396 |
| Traffic 192 | | 0.442±0.001 | 0.289±0.001 | 0.457 | 0.294 | 0.426 | 0.290 | 0.452 | 0.307 | **0.417** | **0.276** | 0.454 | 0.289 | 0.602 | 0.363 | 0.595 | 0.377 | 0.598 | 0.370 |
| Traffic 336 | | 0.456±0.002 | 0.295±0.002 | 0.470 | 0.299 | 0.432 | 0.281 | 0.470 | 0.316 | 0.433 | 0.283 | 0.454 | 0.282 | 0.609 | 0.366 | 0.609 | 0.385 | 0.605 | 0.373 |
| Traffic 720 | | 0.487±0.003 | 0.311±0.001 | 0.502 | 0.314 | 0.463 | **0.300** | 0.498 | 0.323 | 0.467 | 0.302 | 0.484 | 0.301 | 0.647 | 0.385 | 0.673 | 0.418 | 0.645 | 0.394 |
| Traffic avg | | 0.454±0.002 | 0.293±0.001 | 0.472 | 0.301 | 0.431 | 0.287 | 0.463 | 0.310 | **0.428** | **0.282** | 0.454 | 0.286 | 0.627 | 0.376 | 0.618 | 0.390 | 0.625 | 0.383 |

Table 2: Performance comparison of average prediction lengths with lookback lengths $L \in \{336, 720\}$. The best results are highlighted in **bold** and the second best are in underlined.

| Lookback | | L = 336 | | | | | | | | L = 720 | | | | | | | |
|---|---|---|---|---|---|---|---|---|---|---|---|---|---|---|---|---|---|
| Model | | TimeEmb | | CycleNet | | FilterNet | | iTransformer | | TimeEmb | | CycleNet | | SOFTS | | DLinear | |
| Metric | | MSE | MAE | MSE | MAE | MSE | MAE | MSE | MAE | MSE | MAE | MSE | MAE | MSE | MAE | MSE | MAE |
| ETTh1 | | **0.410** | **0.423** | 0.415 | 0.426 | 0.423 | 0.437 | 0.440 | 0.447 | **0.418** | **0.433** | 0.430 | 0.439 | 0.434 | 0.455 | 0.437 | 0.448 |
| ETTm1 | | **0.340** | **0.371** | 0.355 | 0.379 | 0.352 | 0.381 | 0.365 | 0.392 | **0.345** | **0.376** | 0.355 | 0.381 | 0.364 | 0.396 | 0.367 | 0.391 |
| ETTm2 | | **0.247** | **0.303** | 0.251 | 0.309 | 0.265 | 0.325 | 0.286 | 0.337 | **0.248** | **0.308** | 0.249 | 0.312 | 0.268 | 0.331 | 0.261 | 0.327 |
| Weather | | **0.221** | 0.255 | 0.226 | 0.266 | 0.224 | **0.239** | 0.236 | 0.272 | **0.218** | **0.257** | 0.224 | 0.266 | 0.230 | 0.272 | 0.240 | 0.292 |

58, 14]. Forecasting metrics include MSE and MAE, with results averaged over five independent runs. TimeEmb is trained for 30 epochs with early stopping (patience = 5 on the validation set). Batch sizes are 256 for ETTs and the Weather dataset, and 64 for others. Learning rates are selected from $\{0.0005, 0.001, 0.002, 0.005\}$, with TimeEmb's hidden layer size fixed at 512. Experiments use PyTorch 2.1 [33] on an NVIDIA RTX 4090 24GB GPU, with details in **Appendix C**.

## 4.2 Overall Performance

Table 1 presents the comparison results with $L = 96$ and $H \in \{96, 192, 336, 720\}$. The baseline results are from the original papers. Several conclusions can be made as follows:

(1) ***TimeEmb consistently outperforms strong baselines across diverse datasets.*** Across multiple benchmarks and forecast horizons, TimeEmb achieves a significant reduction in MSE, with relative improvements ranging from 3.0% to 8.7% on average. This highlights the effectiveness of our frequency-based dynamic-static decomposition framework, which explicitly separates and models time-invariant and time-varying components.

(2) ***TimeEmb surpasses disentanglement-based baselines by offering more expressive and flexible decomposition.*** While CycleNet relies on a single long-period embedding and DLinear adopts a local moving average for trend extraction, both approaches struggle to capture long-term temporal patterns effectively. In contrast, TimeEmb leverages a global, timestamp-aware embedding bank to learn and represent recurring invariant patterns, enabling more accurate long-range forecasting.

Table 3: Performance of integrating TimeEmb with different backbones on Electricity and Weather. The best results are **bold**. Impr. indicates the performance improvement by equipping TimeEmb.

| Dataset | Electricity | | | | | | | | Weather | | | | | | | |
|---|---|---|---|---|---|---|---|---|---|---|---|---|---|---|---|---|
| Horizon | 96 | | 192 | | 336 | | 720 | | 96 | | 192 | | 336 | | 720 | |
| Metric | MSE | MAE | MSE | MAE | MSE | MAE | MSE | MAE | MSE | MAE | MSE | MAE | MSE | MAE | MSE | MAE |
| Linear | 0.196 | 0.279 | 0.195 | 0.282 | 0.208 | 0.298 | 0.243 | 0.330 | 0.197 | 0.256 | 0.238 | 0.295 | 0.285 | 0.335 | **0.346** | 0.381 |
| + our model | **0.173** | **0.270** | **0.179** | **0.274** | **0.193** | **0.288** | **0.233** | **0.320** | **0.170** | **0.218** | **0.222** | **0.260** | **0.275** | **0.298** | 0.349 | **0.345** |
| Impr. | +11.7% | +3.2% | +8.2% | +2.8% | +7.2% | +3.4% | +4.1% | +3.0% | +13.7% | +14.8% | +6.7% | +11.9% | +3.5% | +11.0% | -0.9% | +9.4% |
| MLP | 0.177 | 0.265 | 0.183 | 0.271 | 0.197 | 0.287 | 0.234 | 0.320 | 0.180 | 0.234 | 0.223 | 0.274 | 0.268 | 0.309 | **0.342** | 0.370 |
| + our model | **0.137** | **0.234** | **0.155** | **0.250** | **0.172** | **0.267** | **0.211** | **0.303** | **0.154** | **0.197** | **0.203** | **0.243** | **0.263** | **0.288** | 0.344 | **0.344** |
| Impr. | +22.6% | +11.7% | +15.3% | +7.7% | +12.7% | +7.0% | +9.8% | +5.3% | +14.4% | +15.8% | +9.0% | +11.3% | +1.9% | +6.8% | -0.6% | +7.0% |
| DLinear | 0.195 | 0.278 | 0.194 | 0.281 | 0.207 | 0.297 | 0.243 | 0.330 | 0.195 | 0.254 | 0.237 | 0.295 | 0.281 | 0.329 | 0.347 | 0.385 |
| + our model | **0.171** | **0.271** | **0.181** | **0.281** | **0.190** | **0.291** | **0.223** | **0.321** | **0.168** | **0.230** | **0.216** | **0.277** | **0.264** | **0.316** | **0.333** | **0.370** |
| Impr. | +12.3% | +2.5% | +6.7% | +0.0% | +8.2% | +2.0% | +8.2% | +2.7% | +13.8% | +9.4% | +8.9% | +6.1% | +6.0% | +4.0% | +4.0% | +3.9% |
| iTransformer | 0.153 | 0.245 | 0.166 | **0.256** | 0.182 | **0.274** | 0.218 | 0.306 | 0.181 | 0.222 | 0.226 | 0.260 | 0.284 | 0.302 | 0.360 | 0.352 |
| + our model | **0.142** | **0.242** | **0.163** | 0.260 | **0.175** | 0.275 | **0.203** | **0.299** | **0.162** | **0.208** | **0.210** | **0.251** | **0.269** | **0.296** | **0.346** | **0.344** |
| Impr. | +7.2% | +1.2% | +1.8% | -1.6% | +3.8% | -0.4% | +6.9% | +2.3% | +10.5% | +6.3% | +7.1% | +3.5% | +5.3% | +2.0% | +3.9% | +2.3% |

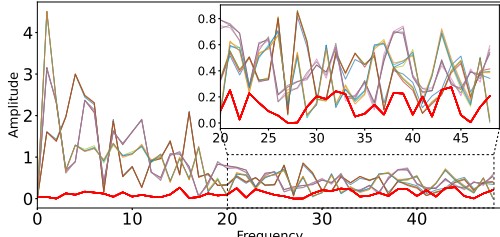

(a) Time series $\overline{\boldsymbol{X}}$ spectrum (colorful lines) and time-invariant embedding $\boldsymbol{X}_s$ spectrum (bold red line)

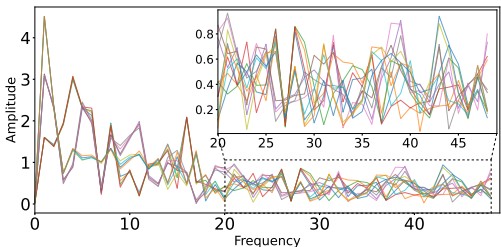

(b) Time-varying component $\boldsymbol{X}_d$ spectrum

Figure 3: Disentangled features visualization in frequency spectrum. Frequency components from 20 to 49 are zoomed in at the top right corner.

(3) ***TimeEmb outperforms frequency-domain models by jointly modeling invariant and dynamic components.*** FilterNet and FITS adopt a fixed filtering approach, which may not effectively handle non-stationary frequency components, as they ignore the global invariant patterns. Conversely, except for the frequency filter for the time-varying component, the embeddings in TimeEmb can preserve long-term invariant patterns, indicating the structural information of time series.

To evaluate model efficiency, we compare the number of trainable parameters, training time, and MSE on the Electricity dataset against mainstream baselines, as shown in Figure 1. Notably, TimeEmb uses over *5× fewer parameters* than the representative Transformer-based model iTransformer, while simultaneously achieving the *best predictive performance*. Benefiting from its lightweight design, TimeEmb significantly accelerates training without compromising accuracy, demonstrating an exceptional balance between efficiency and effectiveness.

To further assess the model's ability to capture long-term dependencies, we evaluate TimeEmb under extended lookback windows. Table 2 reports the average performance across all prediction lengths for $L = 336$ and $L = 720$. Full results are deferred to **Appendix D**. TimeEmb maintains state-of-the-art performance under long input horizons, showcasing its strong temporal modeling capacity.

### 4.3 Compatibility Analysis

To assess the generalizability of our proposed disentanglement mechanism for decoupling time-invariant and time-varying components, we integrate it into several state-of-the-art time series forecasting models, spanning both MLP-based and Transformer-based architectures. Specifically, our method only replaces the backbone prediction layer with the alternative model in the time domain, enabling seamless integration with different models. As shown in Table 3, incorporating our method consistently improves baseline performance across various prediction horizons, validating its effectiveness as a plug-and-play enhancement for diverse forecasting frameworks. Importantly, this integration incurs minimal computational overhead, enabling seamless adoption without significantly increasing model complexity or training cost. These results highlight the broad applicability of our disentanglement framework and its potential to strengthen existing models with negligible trade-offs.

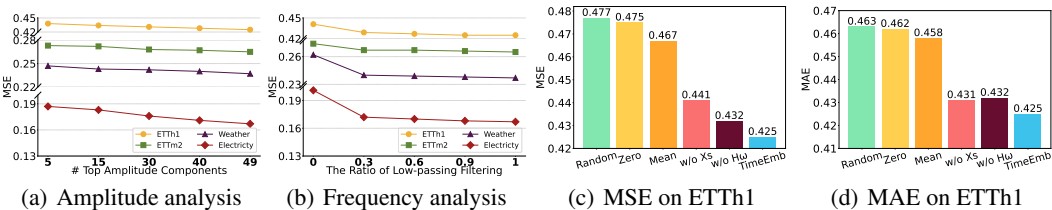

| (a) Amplitude analysis | (b) Frequency analysis | (c) MSE on ETTh1 | (d) MAE on ETTh1 |

Figure 5: Key components contribution analysis.

## 4.4 Disentangled Features Visualization

To evaluate the disentanglement capability of TimeEmb in separating time-invariant and time-varying components, we conduct diverse visualization illustrations. We first select the first channel from the ETTm2 dataset and extract a dozen time series ending at 0 o'clock (*i.e.,* time index $t_{last} = 0$) from different days. In Figure 3(a), we illustrate the frequency-domain representations $\overline{X}$ of these series as colorful lines, and the corresponding learned time-invariant embedding $X_s$ as a bold red line. For clarity, we zoom in on frequency components in the range of 20 to 49. We can observe that the multiple $\overline{X}$ from different days exhibit similar spectral structures, and the learned corresponding time-invariant embedding $X_s$ captures the common pattern to a certain extent. Figure 3 (b) shows the time-varying component $X_d$, which are relatively distinct from one another. The results clearly indicate that the original time series are hard to distinguish, but they become more separable after subtracting the time-invariant embedding. It shows that our TimeEmb successfully captures the shared time-invariant components across the input sequences, preserving the general structural information.

In addition, we present the distribution of the data before and after disentanglement from a high-level perspective.

We project the data samples from the Electricity test set onto a two-dimensional space using the T-SNE [45]. To capture week-level time-invariant patterns, we add an embedding bank composed of 7 learnable embeddings.

The time series $\overline{X}$ and the time-varying component $X_d$ are depicted in Figure 4 (a) and (b), respectively. Time series corresponding to different days of the week are color-coded. For example, the time series for Monday is in dark blue, and that for Sunday is in light yellow. As can be observed from Figure 4 (a), the time series $\overline{X}$ from different days of the week tend to be intermixed, suggesting the existence of inherent similar patterns among them. After separating the time-invariant component from $\overline{X}$, the time-varying components $X_d$ for different days of the week carry specific information, as evidenced by their distinct and isolated distribution in Figure 4 (b). This visualization supports our previous finding again: by disentangling the time-invariant component $X_s$, the time-varying component $X_d$ becomes more distinguishable compared to the original time series $\overline{X}$.

More visualization of TimeEmb can be found in **Appendix D**.

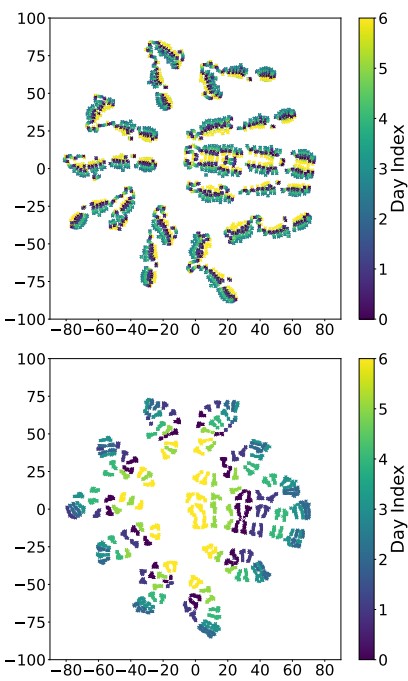

Figure 4: T-SNE visualization results.

## 4.5 Ablation Study

We conduct a comprehensive ablation study to evaluate the contributions of the key components in TimeEmb. The analysis is performed from two main perspectives: (1) the frequency composition of the time-invariant embedding, and (2) the impact of removing or altering individual modules.

### 4.5.1 Frequency Spectrum Analysis of $X_s$

To examine how different frequency components contribute to the time-invariant embedding $X_s$, we design two controlled perturbation strategies. **Amplitude-based masking**: For each input $\overline{X}$, we preserve only the top-$k$ frequency components of $X_s$ with the highest amplitudes, and zero out the rest. **Frequency-based filtering**: We apply a low-pass filter by retaining only a certain proportion of low-frequency components, discarding the high-frequency parts. The results are shown in Figure 5 (a) and (b), with full details provided in **Appendix D**. As shown in Figure 5(a), model performance improves as more high-amplitude components are retained, indicating that both principal and subordinate frequencies carry useful invariant information [52]. Similarly, Figure 5(b) shows that increasing the proportion of low-frequency components leads to better performance, reflecting the importance of capturing both short-term and long-term periodicities in the invariant representation. These findings support the use of the full spectrum in constructing $X_s$.

### 4.5.2 Component-wise Ablation

To assess the individual impact of key modules, we construct several variants of TimeEmb by altering or removing components: **Random**: The embedding bank is randomly initialized between training and testing. **Zero/Mean**: The embedding bank is fixed to zeros or the global mean value, respectively. **w/o $X_s$**: The time-invariant component is entirely removed. **w/o $\mathcal{H}_\omega$**: The frequency filter is removed from the dynamic processing path. The results in Figure 5 (c) and (d) demonstrate that both the embedding bank and the frequency filter substantially contribute to model performance. In particular, removing either module leads to notable degradation, confirming the importance of jointly modeling the time-invariant and time-varying components. Complete ablation results are reported in **Appendix D**. Together, these findings validate the effectiveness of our systematic disentanglement framework, in which $X_s$ and $X_d$ are processed independently via dedicated structures to capture complementary temporal characteristics.

## 5 Conclusion

In this paper, we tackle the crucial issue of temporal non-stationarity in time series forecasting using a well-structured decomposition framework. We introduce TimeEmb, a lightweight yet effective architecture that combines global temporal embeddings and spectral filtering. TimeEmb enables separate processing of the disentangled time-variant and time-invariant components. Specifically, we utilize learnable embeddings to preserve the long-term invariant patterns within time series. Moreover, we devise a frequency filter to capture the temporal dependencies of the time-varying component. Extensive experiments confirm that our method not only attains state-of-the-art performance but also offers interpretable insights into temporal patterns via its dual-path design. It achieves an outstanding balance between performance and efficiency. Furthermore, it can be easily integrated with existing methods, thereby enhancing the ability to predict time series.

## Acknowledgements

This work was supported by the China Postdoctoral Science Foundation under Grant No. 2025M771587; the Open Funding Programs of State Key Laboratory of AI Safety (2025-09); National Science and Technology Major Project under Grant No. 2021ZD0112500; the National Natural Science Foundation of China under Grant Nos. U22A2098, 62172185, 62206105, and 62202200; the Major Science and Technology Development Plan of Jilin Province under Grant No.20240212003GX, the Major Science and Technology Development Plan of Changchun under Grant No.2024WX05; the XJTU Research Fund for AI Science (No. 2025YXYC004).

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

# A  In-Depth Analysis of TimeEmb

## A.1  Innovation Discussions

***TimeEmb vs. Disentanglement Methods.*** While prior disentanglement approaches [31, 51] focus primarily on separating trend and residual components based on local statistics within individual time series, TimeEmb introduces two fundamental advancements. First, instead of local decomposition, our model leverages a learnable embedding bank to capture globally consistent and recurrent patterns across the entire dataset, effectively preserving system-level invariants. Second, TimeEmb is the first to incorporate learnable frequency-domain filtering into the disentanglement framework, enabling efficient and expressive modeling of dynamic components in the spectral space.

***TimeEmb vs. Embedding-enhanced Methods.*** Embedding-enhanced models [40, 39] typically use identifier-based embeddings (*e.g.,* time slot, spatial ID) to encode auxiliary information. In contrast, TimeEmb adopts a decomposition-based design where a learnable temporal embedding bank explicitly models the time-invariant signal component. This enables data-driven recovery of latent periodic patterns without relying on predefined identifiers or external priors.

## A.2  Theoritical Support

In this section, we theoretically analyze the core design of TimeEmb from a frequency-domain perspective. We focus on two main aspects: the completeness of frequency-domain representation and operations, and the expressiveness of the learnable spectral filtering mechanism for modeling dynamic temporal signals.

### A.2.1  Completeness of Frequency-Domain Representation and Operations

TimeEmb operates entirely in the frequency domain by applying the real-valued Fast Fourier Transform (rFFT) to input sequences. For a real-valued time series $X \in \mathbb{R}^{L \times D}$, its spectral representation is obtained as $\overline{X} \in \mathbb{C}^{F \times D}$, where $F = \lfloor L/2 \rfloor + 1$ due to the conjugate symmetry of the spectrum. The rFFT is defined as:

$$\overline{X}[k] = \sum_{n=0}^{L-1} X[n] \cdot e^{-2\pi jkn/L}, \quad k = 0, \dots, F-1. \tag{8}$$

This transformation is invertible via the corresponding inverse real FFT (irFFT), guaranteeing that no information is lost in the process. Thus, rFFT offers a complete and efficient frequency representation of real-valued signals [30, 5]. Beyond transformation, TimeEmb performs a sequence of operations entirely in the frequency domain:

1. Subtraction of a time-invariant embedding $X_s$ from the input spectrum $\overline{X}$;

2. Frequency-wise modulation of the residual $X_d = \overline{X} - X_s$ via a learnable filter $\omega$;

3. Reconstruction of the final spectrum $\dot{X} = X_s + X_d \odot \omega$, followed by an irFFT to recover the output in the time domain.

Each of these operations, *i.e.,* subtraction, modulation, and addition, is algebraically well-defined and closed in the frequency domain. Because the rFFT is invertible, the entire transformation chain in TimeEmb is representation-complete: all original information is preserved, while allowing structured manipulation in the spectral space.

This design offers several important advantages. First, it enables precise modeling of periodicity and oscillatory behavior, which are often hard to localize in the time domain. Second, working entirely in the frequency space allows for efficient and interpretable decomposition of long-range temporal patterns. Lastly, the model avoids any information loss due to projection or truncation, ensuring theoretical soundness in its design.

### A.2.2  Expressiveness of Frequency-Domain Filtering

To model the dynamic (time-varying) component of the input sequence, TimeEmb applies a frequency-domain filter over the residual spectrum. Formally, given the residual $X_d \in \mathbb{C}^{F \times D}$, a learnable

modulation vector $\boldsymbol{\omega} \in \mathbb{C}^{F \times 1}$ is applied as:

$$\mathcal{H}_\omega(\boldsymbol{X}_d)[k] = \boldsymbol{X}_d[k] \odot \boldsymbol{\omega}[k]. \tag{9}$$

This operation is grounded in the **Convolution Theorem**: pointwise multiplication in the frequency domain corresponds to convolution in the time domain [30]. Therefore, the frequency filter can be interpreted as learning the impulse response of a Linear Time-Invariant (LTI) system directly in the spectral domain.

This interpretation grants the model several expressive and practical advantages [3]. First, it enables the learning of flexible signal transformations that go beyond local convolutions, *e.g.,* capturing long-range dependencies with global frequency-aware operations. Second, the filtering process is computationally efficient, operating in $\mathcal{O}(F \times D)$, and avoids the kernel length constraints inherent in time-domain CNNs. Finally, this formulation provides intuitive control over the model's sensitivity to various periodic structures, allowing it to emphasize or suppress spectral bands depending on task-specific dynamics.

In essence, the frequency filter in TimeEmb serves as a powerful and compact operator that simulates a broad family of spectral responses.

### A.3 Convolution Theorem

Frequency filtering modifies a signal's frequency content. Given a signal $x[n]$ and a filter with frequency response $H[k]$, the filtered signal $Y[k] = X[k]H[k]$ in the frequency domain. By the convolution theorem, the filtered signal $y[n] = \text{IDFT}(Y[k]) = (x \circledast h)[n]$ in the time domain. The proof is as follows:

Let $x[n]$ and $h[n]$ be length - $N$ sequences with DFTs $X[k]$ and $H[k]$:

$$X[k] = \sum_{n=0}^{N-1} x[n]e^{-j\frac{2\pi}{N}kn}, \quad k = 0, 1, \cdots, N-1, \tag{10}$$

$$H[k] = \sum_{n=0}^{N-1} h[n]e^{-j\frac{2\pi}{N}kn}, \quad k = 0, 1, \cdots, N-1. \tag{11}$$

The circular convolution of $x[n]$ and $h[n]$ is defined as $y[n] = (x \circledast h)[n] = \sum_{m=0}^{N-1} x[m]h[(n - m) \bmod N]$, where $\circledast$ represents the circular convolution operation, and $(n - m) \bmod N$ denotes the modulo $N$ operation of $n - m$.

The DFT of $y[n]$, denoted as $Y[k]$:

$$Y[k] = \sum_{n=0}^{N-1} y[n]e^{-j\frac{2\pi}{N}kn} \tag{12}$$

$$= \sum_{n=0}^{N-1} \left( \sum_{m=0}^{N-1} x[m]h[(n - m) \bmod N] \right) e^{-j\frac{2\pi}{N}kn}. \tag{13}$$

Let $l = (n - m) \bmod N$, and the above equation can be rewritten as:

$$Y[k] = N \sum_{m=0}^{N-1} x[m] \sum_{l=0}^{N-1} h[l]e^{-j\frac{2\pi}{N}k(l+m)}. \tag{14}$$

According to the exponential operation rule, we have:

$$Y[k] = \left( \sum_{m=0}^{N-1} x[m]e^{-j\frac{2\pi}{N}km} \right) \left( \sum_{l=0}^{N-1} h[l]e^{-j\frac{2\pi}{N}kl} \right). \tag{15}$$

Therefore:

$$Y[k] = X[k]H[k]. \tag{16}$$

**Algorithm 1** Workflow of TimeEmb.

---

**Input**: Time series $\boldsymbol{X} \in \mathbb{R}^{L \times D}$.
**Output**: Prediction $\widehat{\boldsymbol{X}} \in \mathbb{R}^{H \times D}$.

1: // Domain transformation
2: $\overline{\boldsymbol{X}} = \text{FFT}(\text{InstNorm}(\boldsymbol{X}))$
3: // Time series disentanglement
4: $\boldsymbol{X}_d = \overline{\boldsymbol{X}} - \boldsymbol{X}_s$ {Eq. (2)}
5: // Frequency filtering
6: $\dot{\boldsymbol{X}} = \mathcal{H}_{\boldsymbol{\omega}}(\boldsymbol{X}_d) + \boldsymbol{X}_s$ {Eq. (4)}
7: // Final prediction
8: $\widehat{\boldsymbol{X}} = \text{InvNorm}(f_{\boldsymbol{\theta}}(\text{IFFT}(\dot{\boldsymbol{X}})))$ {Eq. (6)}
9: **Return:** $\widehat{\boldsymbol{X}}$

---

Table 4: Dataset statistics. "Channels" denotes the number of variables in each dataset; "M of each bank" denotes the capacity of each embedding bank utilized in TimeEmb. "d" refers to the day-level embedding bank, while "w" indicates the week-level embedding bank.

| Datasets | ETTh1 | ETTh2 | ETTm1 | ETTm2 | Electricity | Weather | Traffic |
|---|---|---|---|---|---|---|---|
| Channels | 7 | 7 | 7 | 7 | 321 | 21 | 862 |
| Timesteps | 17420 | 17420 | 69680 | 69680 | 26304 | 52696 | 17544 |
| Frequency | Hourly | Hourly | 15min | 15min | Hourly | 10min | Hourly |
| Domain | Electricity | Electricity | Electricity | Electricity | Electricity | Weather | Traffic |
| M of each bank | 24 (d) | 24 (d) | 24 (d) | 24 (d) | 24 (d) + 7 (w) | 24 (d) | 24 (d) + 7 (w) |

The same is true for inverse derivation. We can ultimately infer that:

$$y[n] = \text{IDFT}(\mathcal{Y}[k]) = (x \circledast h)[n] = \sum_{m=0}^{N-1} x[m]h[(n-m) \mod N]. \tag{17}$$

In conclusion, we have proved that the DFT of the circular convolution is equal to the product of the DFTs, and the IDFT of the product of the DFTs is equal to the circular convolution, which means that frequency filtering (multiplication in the DFT domain) is equivalent to circular convolution in the time domain.

## B Algorithm

We present the pipeline of TimeEmb in Algorithm 1. We begin by performing a domain transformation for frequency analysis (line 2). Specifically, we apply instance normalization to the input sequence $\boldsymbol{X}$, followed by a Fast Fourier Transformation (FFT) to obtain the frequency series $\overline{\boldsymbol{X}}$. Next, to disentangle the time series, we retrieve the corresponding embedding from the embedding bank, which serves as the time-invariant component $\boldsymbol{X}_s$. We then separate it from $\overline{\boldsymbol{X}}$ to extract the time-varying component $\boldsymbol{X}_d$ (line 4). Subsequently, we implement frequency filtering with a spectral modulation operator $\boldsymbol{\omega}$ to effectively model the dynamic component. After this step, we add back the time-invariant series $\boldsymbol{X}_s$ (line 6). The combined series then undergoes the Inverse Fast Fourier Transform (IFFT), followed by a projection layer, and concludes with inverse normalization to generate the final prediction (line 8).

## C Experimental Setup

### C.1 Datasets

We detail the description of the datasets here:

**ETT (Electricity Transformer Temperature)** contains two subsets of data: ETTh and ETTm. These datasets are based on hourly and 15-minute intervals, collected from electricity transformers between July 2016 and July 2018.

**Weather** records 21 weather features, including air temperature and humidity, every ten minutes throughout 2020.

**Electricity** collects 321 clients' electricity consumption hourly from 2012 to 2014.

**Traffic** comprises the hourly data of 862 sensors of San Francisco freeways from 2015 to 2016.

Detailed statistics are displayed in Table 4.

## C.2 Baselines

We compare TimeEmb with 9 representative and state-of-the-art models to evaluate the performance and effectiveness, including Frequency-based models, MLP-based models, and Transformer-based models. The details of these baselines are as follows:

**FilterNet** proposes two kinds of learnable filters-Plain shaping filter and Contextual shaping filter-to approximately surrogate the linear and attention mappings widely adopted in time series literature. The detailed implementation is available at `https://github.com/aikunyi/FilterNet`.

**FITS** conducts time series analysis using interpolation in the complex frequency domain, achieving low cost with 10K parameters. The detailed implementation is available at `https://github.com/VEWOXIC/FITS`.

**FreTS** presents a new approach to utilizing MLPs in the frequency domain, effectively capturing the underlying patterns of time series while benefiting from a global view and energy compaction. The detailed implementation is available at `https://github.com/aikunyi/FreTS`.

**DLinear** employs a straightforward one-layer linear model to capture temporal relationships through season-trend decomposition. The detailed implementation is available at `https://github.com/cure-lab/LTSF-Linear`.

**SOFTS** introduces an efficient MLP-based model that utilizes a centralized strategy to enhance performance and lessen dependence on the quality of each channel. The detailed implementation is available at `https://github.com/Secilia-Cxy/SOFTS`.

**CycleNet** utilizes an RCF technique to separate the inherent periodic patterns within sequences and then performs predictions on the residual components of the modeled cycles. The detailed implementation is available at `https://github.com/ACAT-SCUT/CycleNet`.

**iTransformer** uses attention and feed-forward networks on inverted dimensions. It embeds time points of individual series into variate tokens for the attention mechanism to capture multivariate correlations. Additionally, the feed-forward network is applied to each variate token to learn non-linear representations. The detailed implementation is available at `https://github.com/thuml/iTransformer`.

**PatchTST** breaks down time series data into subseries-level patches, which helps in extracting local semantic information. The detailed implementation is available at `https://github.com/yuqinie98/PatchTST`.

**Fredformer** is a Transformer-based framework that addresses frequency bias by equally learning features across various frequency bands, which ensures the model does not neglect lower amplitude features that are crucial for accurate forecasting. The detailed implementation is available at `https://github.com/chenzRG/Fredformer`.

## C.3 Implementation Details

We implemented TimeEmb with PyTorch and conducted experiments on a single NVIDIA RTX4090 GPU that has 24GB of memory. TimeEmb was trained for 30 epochs, with early stopping implemented and a patience level of 5 based on the validation set. The batch size was set to 256 for both the ETT and Weather datasets, while a batch size of 64 was used for the remaining datasets. This adjustment was necessary because the latter datasets have a larger number of channels, which requires a smaller batch size to prevent out-of-memory issues. The learning rate was chosen from the range 0.0005, 0.001, 0.002, 0.005, based on the performance on the validation set. The size of the hidden layer in TimeEmb was consistently set to 512.

Table 5: Full results with lookback lengths $L = 336$. The best results are in **bold** and the second best are underlined.

| Model | | TimeEmb | | CycleNet | | FilterNet | | SOFTS | | iTransformer | | DLinear | |
|---|---|---|---|---|---|---|---|---|---|---|---|---|---|
| Metric | | MSE | MAE | MSE | MAE | MSE | MAE | MSE | MAE | MSE | MAE | MSE | MAE |
| ETTh1 | 96 | **0.367** | **0.394** | 0.374 | 0.396 | 0.379 | 0.404 | 0.386 | 0.405 | 0.396 | 0.415 | 0.374 | 0.398 |
| | 192 | **0.403** | **0.414** | 0.406 | 0.415 | 0.417 | 0.428 | 0.428 | 0.432 | 0.434 | 0.438 | 0.430 | 0.440 |
| | 336 | **0.422** | **0.425** | 0.431 | 0.430 | 0.437 | 0.443 | 0.449 | 0.448 | 0.452 | 0.451 | 0.442 | 0.445 |
| | 720 | **0.446** | **0.459** | 0.450 | 0.464 | 0.458 | 0.472 | 0.460 | 0.476 | 0.476 | 0.485 | 0.497 | 0.507 |
| | avg | **0.410** | **0.423** | 0.415 | 0.426 | 0.423 | 0.437 | 0.431 | 0.440 | 0.440 | 0.447 | 0.436 | 0.448 |
| ETTh2 | 96 | **0.276** | **0.333** | 0.279 | 0.341 | 0.302 | 0.356 | 0.298 | 0.356 | 0.334 | 0.379 | 0.281 | 0.347 |
| | 192 | **0.335** | **0.378** | 0.342 | 0.385 | 0.350 | 0.393 | 0.360 | 0.394 | 0.413 | 0.424 | 0.367 | 0.404 |
| | 336 | **0.370** | **0.405** | 0.371 | 0.413 | 0.376 | 0.414 | 0.385 | 0.415 | 0.414 | 0.432 | 0.438 | 0.454 |
| | 720 | **0.396** | **0.433** | 0.426 | 0.451 | 0.414 | 0.444 | 0.449 | 0.463 | 0.433 | 0.454 | 0.598 | 0.549 |
| | avg | **0.344** | **0.387** | 0.355 | 0.398 | 0.361 | 0.402 | 0.373 | 0.407 | 0.399 | 0.422 | 0.421 | 0.439 |
| ETTm1 | 96 | **0.282** | **0.332** | 0.299 | 0.348 | 0.289 | 0.344 | 0.296 | 0.350 | 0.303 | 0.357 | 0.307 | 0.350 |
| | 192 | **0.323** | **0.361** | 0.334 | 0.367 | 0.331 | 0.369 | 0.336 | 0.374 | 0.345 | 0.383 | 0.340 | 0.373 |
| | 336 | **0.353** | **0.380** | 0.368 | 0.386 | 0.364 | 0.389 | 0.371 | 0.396 | 0.375 | 0.397 | 0.377 | 0.397 |
| | 720 | **0.403** | **0.410** | 0.417 | 0.414 | 0.425 | 0.423 | 0.433 | 0.432 | 0.435 | 0.432 | 0.433 | 0.433 |
| | avg | **0.340** | **0.371** | 0.355 | 0.379 | 0.352 | 0.381 | 0.359 | 0.388 | 0.365 | 0.392 | 0.364 | 0.388 |
| ETTm2 | 96 | 0.160 | **0.243** | **0.159** | 0.247 | 0.177 | 0.265 | 0.174 | 0.259 | 0.184 | 0.273 | 0.165 | 0.257 |
| | 192 | 0.218 | **0.283** | **0.214** | 0.286 | 0.232 | 0.304 | 0.240 | 0.307 | 0.262 | 0.322 | 0.227 | 0.307 |
| | 336 | **0.265** | **0.316** | 0.269 | 0.322 | 0.284 | 0.339 | 0.295 | 0.342 | 0.307 | 0.351 | 0.304 | 0.362 |
| | 720 | **0.346** | **0.370** | 0.363 | 0.382 | 0.367 | 0.390 | 0.377 | 0.396 | 0.390 | 0.402 | 0.431 | 0.441 |
| | avg | **0.247** | **0.303** | 0.251 | 0.309 | 0.265 | 0.325 | 0.272 | 0.326 | 0.286 | 0.337 | 0.282 | 0.342 |
| Weather | 96 | **0.144** | 0.189 | 0.148 | 0.200 | 0.150 | **0.183** | 0.160 | 0.209 | 0.163 | 0.213 | 0.174 | 0.235 |
| | 192 | **0.187** | 0.233 | 0.190 | 0.240 | 0.193 | **0.221** | 0.204 | 0.250 | 0.203 | 0.250 | 0.219 | 0.281 |
| | 336 | **0.238** | 0.271 | 0.243 | 0.283 | 0.246 | **0.258** | 0.249 | 0.284 | 0.253 | 0.288 | 0.264 | 0.317 |
| | 720 | 0.315 | 0.326 | 0.322 | 0.339 | **0.308** | **0.295** | 0.324 | 0.335 | 0.326 | 0.338 | 0.324 | 0.363 |
| | avg | **0.221** | 0.255 | 0.226 | 0.266 | 0.224 | **0.239** | 0.234 | 0.270 | 0.236 | 0.272 | 0.245 | 0.299 |
| Electricity | 96 | 0.128 | 0.223 | 0.128 | 0.223 | 0.132 | 0.224 | **0.127** | **0.221** | 0.133 | 0.229 | 0.140 | 0.237 |
| | 192 | 0.146 | 0.240 | 0.144 | **0.237** | **0.143** | **0.237** | 0.148 | 0.242 | 0.156 | 0.251 | 0.153 | 0.250 |
| | 336 | 0.161 | 0.256 | 0.160 | 0.254 | **0.155** | **0.253** | 0.166 | 0.261 | 0.172 | 0.267 | 0.169 | 0.267 |
| | 720 | 0.198 | 0.289 | 0.198 | 0.287 | **0.195** | 0.292 | 0.202 | 0.293 | 0.209 | 0.304 | 0.203 | 0.299 |
| | avg | 0.158 | 0.252 | 0.158 | **0.250** | **0.156** | 0.252 | 0.161 | 0.254 | 0.168 | 0.263 | 0.166 | 0.263 |
| Traffic | 96 | 0.381 | 0.263 | 0.386 | 0.268 | 0.398 | 0.289 | **0.346** | **0.246** | 0.361 | 0.255 | 0.410 | 0.282 |
| | 192 | 0.398 | 0.271 | 0.404 | 0.276 | 0.422 | 0.303 | **0.373** | **0.258** | 0.380 | 0.268 | 0.423 | 0.288 |
| | 336 | 0.411 | 0.278 | 0.416 | 0.281 | 0.437 | 0.312 | **0.385** | **0.265** | 0.389 | 0.273 | 0.436 | 0.296 |
| | 720 | 0.439 | 0.294 | 0.445 | 0.300 | 0.464 | 0.325 | 0.419 | **0.283** | **0.415** | 0.285 | 0.466 | 0.315 |
| | avg | 0.407 | 0.277 | 0.413 | 0.281 | 0.430 | 0.307 | **0.381** | **0.263** | 0.386 | 0.270 | 0.434 | 0.295 |

# D  Detailed Results

## D.1  Full Results with Lookback Window Length L $\in \{336, 720\}$

To evaluate the performance of TimeEmb in modeling long-term temporal dependencies, we further conduct experiments with extended lookback window lengths of 336 and 720. As shown in Table 5 and Table 6, TimeEmb consistently achieves competitive or superior performance across various forecast horizons under these challenging settings. Unlike many baseline models whose performance degrades significantly as the input length increases, TimeEmb maintains stable accuracy, demonstrating strong temporal generalization.

This performance stems from the architectural design of TimeEmb. The time-invariant embedding bank allows the model to effectively summarize recurring structural patterns, regardless of input length. Meanwhile, the frequency-domain filter adaptively emphasizes relevant dynamic components without being constrained by local receptive fields. Together, these modules enable TimeEmb to capture both long-range dependencies and localized variations efficiently.

Overall, these results indicate that TimeEmb is not only effective under standard settings, but also exhibits strong scalability and resilience when applied to long-context forecasting tasks—a desirable property for real-world time series applications.

Table 6: Full results with lookback lengths $L = 720$. The best results are in **bold** and the second best are underlined.

| | | TimeEmb | | CycleNet | | FilterNet | | SOFTS | | iTransformer | | DLinear | |
|---|---|---|---|---|---|---|---|---|---|---|---|---|---|
| Model | Metric | MSE | MAE | MSE | MAE | MSE | MAE | MSE | MAE | MSE | MAE | MSE | MAE |
| ETTh1 | 96 | **0.372** | **0.400** | 0.379 | 0.403 | 0.390 | 0.418 | 0.384 | 0.416 | 0.401 | 0.430 | 0.379 | 0.402 |
| | 192 | **0.413** | 0.427 | 0.416 | **0.425** | 0.424 | 0.439 | 0.423 | 0.442 | 0.434 | 0.452 | 0.419 | 0.429 |
| | 336 | **0.438** | **0.443** | 0.447 | 0.445 | 0.450 | 0.461 | 0.446 | 0.461 | 0.468 | 0.475 | 0.456 | 0.456 |
| | 720 | **0.449** | **0.462** | 0.477 | 0.483 | 0.484 | 0.488 | 0.481 | 0.500 | 0.525 | 0.520 | 0.493 | 0.506 |
| | avg | **0.418** | **0.433** | 0.430 | 0.439 | 0.437 | 0.452 | 0.434 | 0.455 | 0.457 | 0.469 | 0.437 | 0.448 |
| ETTh2 | 96 | 0.290 | 0.348 | **0.271** | **0.337** | 0.297 | 0.357 | 0.295 | 0.357 | 0.306 | 0.369 | 0.309 | 0.373 |
| | 192 | 0.347 | 0.387 | **0.332** | **0.380** | 0.361 | 0.400 | 0.365 | 0.401 | 0.372 | 0.409 | 0.409 | 0.433 |
| | 336 | 0.376 | 0.411 | **0.362** | **0.408** | 0.397 | 0.431 | 0.398 | 0.426 | 0.403 | 0.434 | 0.508 | 0.495 |
| | 720 | **0.399** | **0.439** | 0.415 | 0.449 | 0.435 | 0.460 | 0.448 | 0.473 | 0.434 | 0.464 | 0.851 | 0.653 |
| | avg | 0.353 | 0.396 | **0.345** | **0.394** | 0.373 | 0.412 | 0.377 | 0.414 | 0.379 | 0.419 | 0.519 | 0.489 |
| ETTm1 | 96 | **0.293** | **0.346** | 0.307 | 0.353 | 0.301 | 0.358 | 0.299 | 0.357 | 0.317 | 0.367 | 0.309 | 0.353 |
| | 192 | **0.326** | **0.366** | 0.337 | 0.371 | 0.340 | 0.379 | 0.342 | 0.381 | 0.347 | 0.385 | 0.345 | 0.376 |
| | 336 | **0.356** | **0.382** | 0.364 | 0.387 | 0.375 | 0.398 | 0.375 | 0.401 | 0.377 | 0.402 | 0.376 | 0.398 |
| | 720 | **0.405** | **0.410** | 0.410 | 0.411 | 0.434 | 0.426 | 0.441 | 0.443 | 0.429 | 0.431 | 0.436 | 0.436 |
| | avg | **0.345** | **0.376** | 0.355 | 0.381 | 0.363 | 0.390 | 0.364 | 0.396 | 0.368 | 0.396 | 0.367 | 0.391 |
| ETTm2 | 96 | 0.163 | 0.251 | **0.159** | **0.249** | 0.180 | 0.271 | 0.181 | 0.272 | 0.187 | 0.278 | 0.163 | 0.256 |
| | 192 | 0.219 | 0.290 | **0.214** | **0.289** | 0.239 | 0.313 | 0.234 | 0.310 | 0.251 | 0.319 | 0.220 | 0.300 |
| | 336 | **0.265** | **0.320** | 0.268 | 0.326 | 0.283 | 0.341 | 0.284 | 0.342 | 0.307 | 0.355 | 0.283 | 0.347 |
| | 720 | **0.343** | **0.372** | 0.353 | 0.384 | 0.361 | 0.394 | 0.373 | 0.398 | 0.391 | 0.411 | 0.376 | 0.406 |
| | avg | **0.248** | **0.308** | 0.249 | 0.312 | 0.266 | 0.330 | 0.268 | 0.331 | 0.284 | 0.341 | 0.261 | 0.327 |
| Weather | 96 | **0.143** | **0.193** | 0.149 | 0.203 | 0.153 | 0.208 | 0.152 | 0.205 | 0.168 | 0.222 | 0.169 | 0.227 |
| | 192 | **0.188** | **0.237** | 0.192 | 0.244 | 0.199 | 0.250 | 0.199 | 0.251 | 0.209 | 0.256 | 0.213 | 0.271 |
| | 336 | **0.236** | **0.274** | 0.242 | 0.283 | 0.248 | 0.287 | 0.248 | 0.288 | 0.267 | 0.302 | 0.259 | 0.311 |
| | 720 | **0.306** | **0.325** | 0.312 | 0.333 | 0.313 | 0.333 | 0.322 | 0.343 | 0.337 | 0.352 | 0.319 | 0.359 |
| | avg | **0.218** | **0.257** | 0.224 | 0.266 | 0.228 | 0.270 | 0.230 | 0.272 | 0.245 | 0.283 | 0.240 | 0.292 |
| Electricity | 96 | 0.129 | 0.225 | **0.128** | **0.223** | 0.137 | 0.235 | 0.137 | 0.232 | 0.142 | 0.243 | 0.134 | 0.232 |
| | 192 | 0.145 | 0.241 | **0.143** | **0.237** | 0.160 | 0.259 | 0.157 | 0.252 | 0.160 | 0.261 | 0.148 | 0.245 |
| | 336 | 0.161 | 0.257 | **0.159** | **0.254** | 0.174 | 0.274 | 0.172 | 0.268 | 0.179 | 0.281 | 0.163 | 0.263 |
| | 720 | **0.197** | 0.289 | **0.197** | **0.287** | 0.212 | 0.307 | 0.198 | 0.291 | 0.220 | 0.316 | 0.198 | 0.296 |
| | avg | 0.158 | 0.253 | **0.157** | **0.250** | 0.171 | 0.269 | 0.166 | 0.261 | 0.175 | 0.275 | 0.161 | 0.259 |
| Traffic | 96 | 0.374 | 0.268 | 0.374 | 0.268 | 0.386 | 0.285 | **0.355** | **0.253** | 0.358 | 0.254 | 0.388 | 0.275 |
| | 192 | 0.387 | 0.268 | 0.390 | 0.275 | 0.401 | 0.281 | **0.369** | **0.261** | 0.375 | 0.263 | 0.399 | 0.279 |
| | 336 | 0.401 | 0.275 | 0.405 | 0.282 | 0.408 | 0.288 | **0.387** | **0.271** | **0.387** | 0.273 | 0.414 | 0.291 |
| | 720 | 0.434 | 0.292 | 0.441 | 0.302 | 0.447 | 0.306 | **0.409** | **0.286** | 0.418 | 0.292 | 0.449 | 0.308 |
| | avg | 0.399 | 0.274 | 0.403 | 0.282 | 0.411 | 0.290 | **0.380** | **0.268** | 0.385 | 0.271 | 0.413 | 0.288 |

## D.2 Full Results of Efficiency Analysis

To verify the lightweight characteristics of TimeEmb, we conduct several efficiency experiments on it. We first compare the maximum memory(MB), training time, and MSE on the ETTm1 dataset against mainstream baselines. Subsequently, we calculate the extra parameters from TimeEmb based on the compatibility study, which can also prove the extensibility and efficiency of TimeEmb. The detailed results are displayed in Table 7 and Table 8. It indicates that TimeEmb can operate in a resource-constrained environment and achieve excellent performance. Moreover, TimeEmb can play the role of a plug-in module in other models, improving their performance at a low cost.

## D.3 Full Results of Ablation Study

### D.3.1 Ablation Results of Embedding Frequency Spectrum Analysis

The complete results of our embedding frequency spectrum ablation are displayed in Table 9 and Table 10. The term "k" in Table 9 represents the number of frequency components of $X_s$ selected based on their top amplitudes, while the term "$\gamma$" in Table 10 refers to the ratio of low-passing filtering applied to the embeddings. The results reveal that leveraging the entire frequency spectrum leads to the best performance. Note that the more frequency components are covered, the better performance TimeEmb can achieve, which indicates that each frequency in the band is important.

Table 7: Efficiency comparison on ETTm1 dataset shows TimeEmb leads in performance and efficiency.

| Model | Training Time(s/epoch) | MSE | Max Memory(MB) |
|---|---|---|---|
| CycleNet | 1.77 | 0.447 | 91.37 |
| Fredformer | 4.46 | 0.453 | 1512.32 |
| FilterNet | 1.63 | 0.456 | **79.51** |
| iTransformer | 2.44 | 0.482 | 275.12 |
| SOFTS | 2.00 | 0.466 | 183.95 |
| TimeEmb | **1.61** | **0.435** | 82.36 |

Table 8: We equip Fredformer and CycleNet with TimeEmb on ETTh2. It brings stable performance(MSE) gains with trivial extra training costs.

| Horizon | 96 | 192 | 336 | 720 |
|---|---|---|---|---|
| Fredformer | 0.293 | 0.371 | 0.382 | 0.415 |
| +TimeEmb | **0.289** | **0.358** | **0.360** | **0.387** |
| Impr. | 1.4% | 3.5% | 5.8% | 6.7% |
| former param | 32820131 | 33465731 | 9894911 | 13656959 |
| current param | 32830860 | 33476460 | 9905640 | 13667688 |
| extra param | **10729 (0.03%-0.11%)** | | | |
| CycleNet | 0.285 | 0.373 | 0.421 | 0.453 |
| +TimeEmb | **0.277** | **0.351** | **0.399** | **0.415** |
| Impr. | 2.8% | 5.9% | 5.2% | 8.4% |
| former param | 99080 | 148328 | 222200 | 419192 |
| current param | 109809 | 159057 | 232929 | 429921 |
| extra param | **10729 (2.56%-10.83%)** | | | |

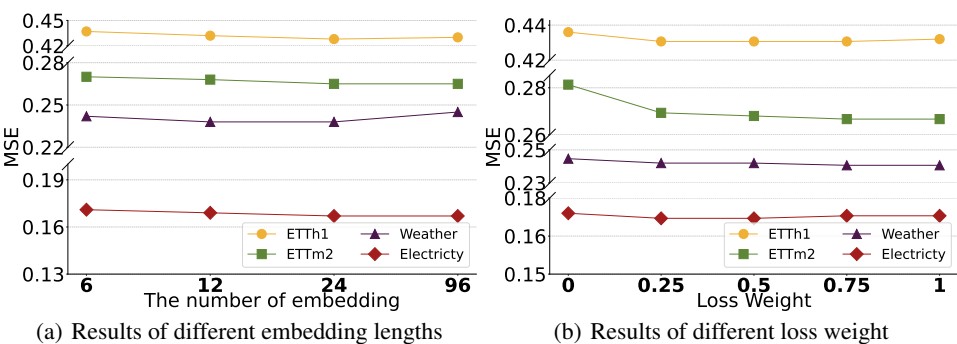

(a) Results of different embedding lengths     (b) Results of different loss weight

Figure 6: Hyper-parameter analysis.

### D.3.2   Ablation Results of Key Components in TimeEmb

In this section, we create different versions of the model by altering or removing specific components from TimeEmb. The results are available in Table 11. **Random**: The embedding bank is randomly initialized between training and testing. **Zero/Mean**: The embedding bank is fixed to zeros or the global mean value, respectively. **"w/o $X_s$"** represents removing the time-invariant embedding $X_s$. **"w/o $\mathcal{H}_\omega$"** indicates removing frequency filter $\mathcal{H}_\omega$. **"w/o RevIN"** refers to removing the reversible instance normalization. It is observed that the time-invariant component embedding contributes most in TimeEmb.

Table 9: Ablation study results. $\boldsymbol{E}$ contains frequency components with top-k amplitude. The best results are in **bold**.

| k | | 5 | | 15 | | 30 | | 40 | | 49(full) | |
|---|---|---|---|---|---|---|---|---|---|---|---|
| Metric | | MSE | MAE | MSE | MAE | MSE | MAE | MSE | MAE | MSE | MAE |
| ETTh1 | 96 | 0.375 | 0.393 | 0.375 | 0.392 | 0.374 | 0.392 | 0.371 | 0.390 | **0.366** | **0.387** |
| | 192 | 0.427 | 0.422 | 0.425 | 0.421 | 0.424 | 0.420 | 0.422 | 0.419 | **0.417** | **0.416** |
| | 336 | 0.465 | 0.441 | 0.463 | 0.440 | 0.463 | 0.440 | 0.459 | 0.439 | **0.457** | **0.436** |
| | 720 | 0.486 | 0.473 | 0.472 | 0.467 | 0.463 | 0.463 | 0.460 | 0.461 | **0.459** | **0.460** |
| | avg | 0.438 | 0.432 | 0.434 | 0.430 | 0.431 | 0.429 | 0.428 | 0.427 | **0.425** | **0.425** |
| ETTm2 | 96 | 0.167 | 0.247 | 0.166 | 0.247 | 0.166 | 0.247 | 0.165 | 0.244 | **0.163** | **0.242** |
| | 192 | 0.238 | 0.295 | 0.236 | 0.293 | 0.230 | 0.288 | 0.230 | 0.287 | **0.226** | **0.285** |
| | 336 | 0.293 | 0.330 | 0.292 | 0.328 | 0.288 | 0.326 | 0.287 | 0.324 | **0.286** | **0.324** |
| | 720 | 0.393 | 0.388 | 0.393 | 0.387 | 0.386 | 0.384 | 0.385 | 0.383 | **0.383** | **0.381** |
| | avg | 0.273 | 0.315 | 0.272 | 0.314 | 0.268 | 0.311 | 0.267 | 0.310 | **0.265** | **0.308** |
| Weather | 96 | 0.158 | 0.202 | 0.155 | 0.197 | 0.154 | 0.195 | 0.152 | 0.192 | **0.150** | **0.190** |
| | 192 | 0.212 | 0.250 | 0.206 | 0.244 | 0.205 | 0.242 | 0.203 | 0.240 | **0.200** | **0.238** |
| | 336 | 0.268 | 0.291 | 0.265 | 0.288 | 0.264 | 0.287 | 0.261 | 0.284 | **0.259** | **0.282** |
| | 720 | 0.350 | 0.344 | 0.347 | 0.342 | 0.346 | 0.341 | 0.344 | 0.340 | **0.339** | **0.336** |
| | avg | 0.247 | 0.272 | 0.243 | 0.268 | 0.242 | 0.266 | 0.240 | 0.264 | **0.237** | **0.262** |
| Electricity | 96 | 0.157 | 0.250 | 0.151 | 0.247 | 0.145 | 0.241 | 0.140 | 0.236 | **0.136** | **0.231** |
| | 192 | 0.171 | 0.261 | 0.168 | 0.261 | 0.161 | 0.255 | 0.156 | 0.250 | **0.153** | **0.246** |
| | 336 | 0.188 | 0.278 | 0.185 | 0.278 | 0.179 | 0.273 | 0.173 | 0.267 | **0.170** | **0.264** |
| | 720 | 0.231 | 0.315 | 0.229 | 0.316 | 0.220 | 0.307 | 0.214 | 0.302 | **0.208** | **0.297** |
| | avg | 0.187 | 0.276 | 0.183 | 0.276 | 0.176 | 0.269 | 0.171 | 0.264 | **0.167** | **0.260** |

Table 10: Ablation study results. $\boldsymbol{E}$ contains frequency components filtered with low-passing filtering by $\gamma$. The best results are in **bold**.

| $\gamma$ | | 0 | | 0.3 | | 0.6 | | 0.9 | | 1 | |
|---|---|---|---|---|---|---|---|---|---|---|---|
| Metric | | MSE | MAE | MSE | MAE | MSE | MAE | MSE | MAE | MSE | MAE |
| ETTh1 | 96 | 0.376 | 0.391 | 0.370 | 0.390 | 0.368 | 0.389 | 0.367 | 0.388 | **0.366** | **0.387** |
| | 192 | 0.431 | 0.422 | 0.420 | 0.418 | 0.420 | 0.417 | 0.418 | 0.417 | **0.417** | **0.416** |
| | 336 | 0.470 | 0.441 | 0.459 | **0.437** | 0.457 | **0.437** | 0.457 | **0.437** | 0.457 | 0.436 |
| | 720 | 0.487 | 0.471 | 0.465 | 0.463 | 0.461 | 0.461 | **0.459** | 0.461 | 0.459 | 0.460 |
| | avg | 0.441 | 0.431 | 0.429 | 0.427 | 0.427 | 0.426 | **0.425** | 0.426 | 0.425 | 0.425 |
| ETTm2 | 96 | 0.173 | 0.252 | 0.164 | 0.243 | 0.164 | 0.243 | 0.164 | 0.243 | **0.163** | **0.242** |
| | 192 | 0.237 | 0.294 | 0.227 | **0.285** | 0.228 | 0.286 | 0.227 | **0.285** | **0.226** | **0.285** |
| | 336 | 0.295 | 0.332 | 0.286 | 0.324 | 0.287 | 0.324 | 0.286 | 0.324 | **0.286** | **0.324** |
| | 720 | 0.391 | 0.389 | **0.382** | 0.382 | 0.383 | 0.382 | 0.383 | 0.382 | 0.383 | **0.381** |
| | avg | 0.274 | 0.317 | 0.267 | 0.309 | 0.267 | 0.309 | 0.266 | 0.309 | **0.265** | **0.308** |
| Weather | 96 | 0.182 | 0.221 | 0.151 | 0.191 | 0.151 | 0.191 | **0.150** | **0.190** | 0.150 | 0.190 |
| | 192 | 0.228 | 0.259 | 0.202 | 0.240 | 0.201 | 0.239 | 0.201 | **0.238** | **0.200** | 0.238 |
| | 336 | 0.282 | 0.299 | 0.261 | 0.284 | 0.260 | 0.283 | **0.259** | 0.283 | 0.259 | 0.282 |
| | 720 | 0.356 | 0.347 | 0.344 | 0.340 | 0.343 | 0.339 | 0.343 | 0.339 | **0.339** | **0.336** |
| | avg | 0.262 | 0.282 | 0.240 | 0.264 | 0.239 | 0.263 | 0.238 | 0.263 | **0.237** | **0.262** |
| Electricity | 96 | 0.178 | 0.259 | 0.141 | 0.236 | 0.139 | 0.234 | 0.138 | 0.233 | **0.136** | **0.231** |
| | 192 | 0.184 | 0.266 | 0.158 | 0.251 | 0.156 | 0.249 | 0.154 | 0.248 | **0.153** | **0.246** |
| | 336 | 0.200 | 0.282 | 0.175 | 0.268 | 0.172 | 0.266 | 0.171 | 0.265 | **0.170** | **0.264** |
| | 720 | 0.241 | 0.316 | 0.214 | 0.301 | 0.211 | 0.299 | 0.210 | 0.298 | **0.208** | **0.297** |
| | avg | 0.201 | 0.281 | 0.172 | 0.264 | 0.170 | 0.262 | 0.168 | 0.261 | **0.167** | **0.260** |

## D.4 Hyper-Parameter Analysis

We conduct experiments to evaluate the impact of essential hyper-parameters of TimeEmb in prediction performance, including the number of embeddings $M$ and loss weight $\alpha$. The hyper-parameters are adjusted individually for each setting based on the performance displayed in Figure 6. Full results can be referred to Table 12 and Table 13.

For the number of embeddings of embedding bank $\boldsymbol{E}$, we set $M \in \{6, 12, 24, 96\}$, and the results are displayed in Figure 6. From the results, it can be observed that, (1) The variation in $M$ settings affects model performance slightly, highlighting the TimeEmb's robustness and its ability to perform well with minimal manual tuning, making it both user-friendly and easy to deploy. (2) Different datasets exhibit distinct characteristics due to their different periodic patterns. For dataset ETTm2, the best performance is achieved when $M = 96$, as the shorter time intervals require a finer granularity of

Table 11: Ablation study results of key modules. The best results are in **bold**.

| Dataset | | ETTh1 | | ETTm2 | | Weather | | Electricity | |
|---|---|---|---|---|---|---|---|---|---|
| Metric | | MSE | MAE | MSE | MAE | MSE | MAE | MSE | MAE |
| TimeEmb | 96 | **0.366** | **0.387** | **0.163** | **0.242** | 0.150 | **0.190** | **0.136** | **0.231** |
| | 192 | **0.417** | **0.416** | **0.226** | **0.285** | 0.200 | **0.238** | **0.153** | **0.246** |
| | 336 | 0.457 | **0.436** | **0.286** | **0.324** | 0.259 | **0.282** | **0.170** | **0.264** |
| | 720 | **0.459** | **0.460** | **0.383** | **0.381** | 0.339 | **0.336** | **0.208** | **0.297** |
| | avg | **0.425** | **0.425** | **0.265** | **0.308** | 0.237 | **0.262** | **0.167** | **0.260** |
| Random | 96 | 0.407 | 0.415 | 0.240 | 0.318 | 0.197 | 0.237 | 0.242 | 0.344 |
| | 192 | 0.453 | 0.443 | 0.299 | 0.351 | 0.249 | 0.279 | 0.251 | 0.354 |
| | 336 | 0.499 | 0.471 | 0.361 | 0.386 | 0.314 | 0.321 | 0.291 | 0.393 |
| | 720 | 0.550 | 0.524 | 0.452 | 0.434 | 0.422 | 0.384 | 0.370 | 0.455 |
| | avg | 0.477 | 0.463 | 0.338 | 0.372 | 0.296 | 0.305 | 0.289 | 0.387 |
| Zero | 96 | 0.405 | 0.414 | 0.240 | 0.317 | 0.198 | 0.237 | 0.239 | 0.342 |
| | 192 | 0.452 | 0.442 | 0.299 | 0.351 | 0.248 | 0.279 | 0.248 | 0.352 |
| | 336 | 0.497 | 0.470 | 0.360 | 0.385 | 0.314 | 0.321 | 0.289 | 0.391 |
| | 720 | 0.547 | 0.523 | 0.450 | 0.433 | 0.421 | 0.383 | 0.367 | 0.453 |
| | avg | 0.475 | 0.462 | 0.337 | 0.372 | 0.295 | 0.305 | 0.286 | 0.385 |
| Mean | 96 | 0.397 | 0.410 | 0.204 | 0.289 | 0.175 | 0.221 | 0.241 | 0.339 |
| | 192 | 0.444 | 0.439 | 0.266 | 0.327 | 0.220 | 0.261 | 0.250 | 0.349 |
| | 336 | 0.488 | 0.464 | 0.328 | 0.364 | 0.267 | 0.296 | 0.291 | 0.387 |
| | 720 | 0.537 | 0.517 | 0.427 | 0.418 | 0.346 | 0.348 | 0.374 | 0.453 |
| | avg | 0.467 | 0.458 | 0.306 | 0.350 | 0.252 | 0.282 | 0.289 | 0.382 |
| w/o. $X_s$ | 96 | 0.376 | 0.391 | 0.173 | 0.252 | 0.182 | 0.221 | 0.178 | 0.259 |
| | 192 | 0.431 | 0.422 | 0.237 | 0.294 | 0.228 | 0.259 | 0.184 | 0.266 |
| | 336 | 0.470 | 0.441 | 0.295 | 0.332 | 0.282 | 0.299 | 0.200 | 0.282 |
| | 720 | 0.487 | 0.471 | 0.391 | 0.389 | 0.356 | 0.347 | 0.241 | 0.316 |
| | avg | 0.441 | 0.431 | 0.274 | 0.317 | 0.262 | 0.282 | 0.201 | 0.281 |
| w/o. $\mathcal{H}_\omega$ | 96 | **0.366** | 0.391 | 0.164 | 0.244 | 0.151 | 0.192 | 0.137 | 0.233 |
| | 192 | **0.417** | 0.421 | 0.229 | 0.287 | 0.201 | 0.239 | 0.155 | 0.249 |
| | 336 | **0.451** | 0.441 | **0.286** | **0.324** | 0.259 | 0.283 | 0.172 | 0.269 |
| | 720 | 0.492 | 0.474 | 0.385 | 0.382 | 0.341 | 0.337 | 0.210 | 0.299 |
| | avg | 0.432 | 0.432 | 0.266 | 0.309 | 0.238 | 0.263 | 0.169 | 0.263 |
| w/o. RevIN | 96 | 0.383 | 0.403 | 0.193 | 0.286 | **0.147** | 0.191 | 0.137 | 0.235 |
| | 192 | 0.450 | 0.449 | 0.283 | 0.352 | **0.194** | **0.238** | 0.154 | 0.252 |
| | 336 | 0.485 | 0.463 | 0.427 | 0.449 | **0.248** | 0.290 | 0.171 | 0.271 |
| | 720 | 0.517 | 0.511 | 0.516 | 0.482 | **0.326** | 0.346 | 0.213 | 0.306 |
| | avg | 0.459 | 0.457 | 0.355 | 0.392 | **0.229** | 0.266 | 0.169 | 0.266 |

embedding to capture temporal patterns. For datasets ETTh1 and Weather, $M = 24$ yields optimal results, effectively capturing data complexity while preventing overfitting.

For the loss weight, we conduct experiments with $\alpha \in \{0, 0.25, 0.5, 0.75, 1\}$. Results in Figure 6 show that an optimal value of $\alpha$ improves TimeEmb's performance. Combining both time domain and frequency domain losses outperforms using time domain loss alone ($\alpha = 0$), highlighting that integrating information from both domains enhances TimeEmb's ability to capture diverse patterns in the time series data.

We examine the impact of several key hyperparameters on TimeEmb 's performance: the number of embeddings in the embedding bank, denoted as "M", and the loss weight in the optimization objective, denoted as "$\alpha$". The detailed results are presented in Table 12 and Table 13. It demonstrates that appropriate hyper-parameters can enhance the performance of TimeEmb.

### D.5 Visualization of the Learned Embeddings

We present the time-invariant component embeddings in Figure 7. The "x" in the term "hour x" represents the hour-index of the last timestep of the input series. Figure 7 depicts the distinct embeddings learned from various datasets and channels. For instance, Figure 7 (a) presents the time-invariant components learned in channel 302 of the electricity dataset, where the hour index is 5. In contrast, Figure 7 (b) shows the time-invariant components in channel 15 of the Weather dataset corresponding to the input series, with the hour index being 2. These embeddings, derived from the global sequence, capture the time-invariant components, offering vital supplementary information to the model and enabling a better understanding of the stable patterns within the time series data.

Table 12: Influence of the number of time-invariant embedding $M$. The best results are in **bold**.

| $M$ | | 6 | | 12 | | 24 | | 96 | |
|---|---|---|---|---|---|---|---|---|---|
| Metric | | MSE | MAE | MSE | MAE | MSE | MAE | MSE | MAE |
| ETTh1 | 96 | 0.372 | 0.389 | 0.370 | 0.388 | **0.366** | **0.387** | 0.371 | 0.390 |
| | 192 | 0.424 | 0.419 | 0.422 | 0.417 | **0.417** | **0.416** | 0.420 | 0.417 |
| | 336 | 0.464 | 0.437 | 0.461 | 0.436 | **0.457** | **0.436** | 0.458 | 0.437 |
| | 720 | 0.481 | 0.465 | 0.464 | 0.462 | **0.459** | **0.460** | 0.460 | 0.461 |
| | avg | 0.435 | 0.428 | 0.429 | 0.426 | **0.425** | **0.425** | 0.427 | 0.426 |
| ETTm2 | 96 | 0.168 | 0.248 | 0.167 | 0.246 | 0.164 | 0.243 | **0.163** | **0.242** |
| | 192 | 0.232 | 0.290 | 0.231 | 0.289 | 0.227 | **0.285** | **0.226** | **0.285** |
| | 336 | 0.291 | 0.328 | 0.290 | 0.327 | **0.285** | **0.323** | 0.286 | 0.324 |
| | 720 | 0.387 | 0.385 | 0.385 | 0.383 | **0.383** | 0.382 | **0.383** | **0.381** |
| | avg | 0.270 | 0.313 | 0.268 | 0.311 | **0.265** | **0.308** | **0.265** | **0.308** |
| Weather | 96 | 0.159 | 0.199 | 0.153 | 0.193 | **0.150** | **0.190** | 0.157 | 0.198 |
| | 192 | 0.205 | 0.242 | 0.202 | 0.239 | **0.201** | **0.238** | 0.209 | 0.246 |
| | 336 | 0.262 | 0.284 | **0.259** | **0.282** | 0.259 | 0.282 | 0.266 | 0.288 |
| | 720 | 0.342 | 0.338 | **0.339** | **0.336** | 0.342 | 0.339 | 0.349 | 0.344 |
| | avg | 0.242 | 0.266 | **0.238** | 0.263 | **0.238** | **0.262** | 0.245 | 0.269 |
| Electricity | 96 | 0.140 | 0.235 | 0.138 | 0.234 | **0.136** | **0.231** | **0.136** | **0.231** |
| | 192 | 0.157 | 0.249 | 0.155 | 0.248 | **0.153** | **0.246** | **0.153** | 0.247 |
| | 336 | 0.173 | 0.266 | 0.172 | 0.266 | **0.170** | **0.264** | **0.170** | **0.264** |
| | 720 | 0.212 | 0.299 | 0.212 | 0.300 | **0.208** | 0.297 | **0.208** | **0.296** |
| | avg | 0.171 | 0.262 | 0.169 | 0.262 | **0.167** | **0.260** | **0.167** | **0.260** |

Table 13: Influence of loss weight $\alpha$. The best results are in **bold**.

| $\alpha$ | | 0 | | 0.25 | | 0.5 | | 0.75 | | 1 | |
|---|---|---|---|---|---|---|---|---|---|---|---|
| Metric | | MSE | MAE | MSE | MAE | MSE | MAE | MSE | MAE | MSE | MAE |
| ETTh1 | 96 | 0.382 | 0.402 | **0.366** | 0.389 | **0.366** | 0.388 | **0.366** | **0.387** | 0.367 | **0.387** |
| | 192 | 0.423 | 0.424 | 0.418 | 0.419 | 0.418 | 0.417 | **0.417** | **0.416** | 0.417 | 0.416 |
| | 336 | 0.463 | 0.444 | 0.463 | 0.440 | 0.460 | 0.438 | **0.457** | 0.437 | 0.457 | **0.436** |
| | 720 | **0.459** | 0.460 | 0.464 | **0.457** | 0.468 | 0.462 | 0.472 | 0.464 | 0.474 | 0.465 |
| | avg | 0.432 | 0.433 | **0.428** | **0.426** | **0.428** | **0.426** | **0.428** | **0.426** | 0.429 | **0.426** |
| ETTm2 | 96 | 0.170 | 0.251 | 0.166 | 0.245 | 0.165 | 0.244 | **0.164** | **0.243** | **0.164** | **0.243** |
| | 192 | 0.234 | 0.293 | 0.230 | 0.288 | 0.228 | 0.286 | **0.227** | **0.285** | 0.227 | 0.285 |
| | 336 | 0.294 | 0.333 | **0.285** | 0.325 | 0.286 | 0.324 | **0.285** | **0.323** | 0.287 | 0.324 |
| | 720 | 0.405 | 0.398 | 0.386 | 0.386 | 0.386 | 0.384 | **0.383** | **0.382** | **0.383** | **0.382** |
| | avg | 0.276 | 0.319 | 0.267 | 0.311 | 0.266 | 0.310 | **0.265** | **0.308** | 0.265 | 0.309 |
| Weather | 96 | 0.154 | 0.197 | 0.151 | 0.193 | 0.151 | 0.193 | **0.150** | **0.190** | **0.150** | **0.190** |
| | 192 | 0.203 | 0.243 | 0.202 | 0.240 | 0.201 | 0.239 | **0.201** | **0.238** | **0.201** | **0.238** |
| | 336 | 0.263 | 0.288 | 0.260 | 0.285 | 0.260 | 0.283 | **0.259** | 0.283 | **0.259** | **0.282** |
| | 720 | 0.344 | 0.344 | **0.342** | 0.340 | **0.342** | 0.340 | **0.342** | **0.339** | **0.342** | **0.339** |
| | avg | 0.241 | 0.268 | 0.239 | 0.265 | 0.239 | 0.264 | **0.238** | 0.263 | **0.238** | **0.262** |
| Electricity | 96 | 0.137 | 0.234 | **0.136** | **0.231** | **0.136** | **0.231** | 0.137 | **0.231** | 0.137 | 0.232 |
| | 192 | 0.155 | 0.250 | **0.153** | **0.246** | **0.153** | **0.246** | 0.154 | 0.247 | 0.154 | 0.247 |
| | 336 | 0.172 | 0.267 | **0.170** | **0.264** | **0.170** | **0.264** | 0.171 | **0.264** | 0.171 | **0.264** |
| | 720 | 0.211 | 0.303 | **0.208** | **0.297** | 0.209 | **0.297** | 0.210 | 0.298 | 0.210 | 0.298 |
| | avg | 0.169 | 0.264 | **0.167** | **0.260** | **0.167** | **0.260** | 0.168 | 0.260 | 0.168 | 0.260 |

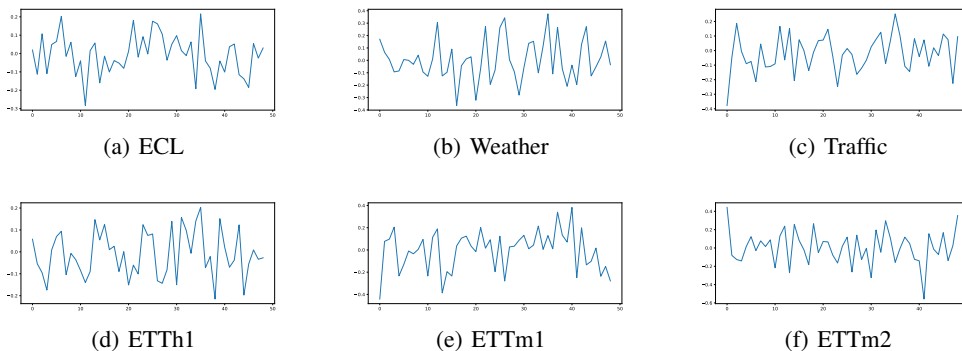

(a) ECL    (b) Weather    (c) Traffic

(d) ETTh1    (e) ETTm1    (f) ETTm2

Figure 7: Visualization of the learned time-invariant embeddings $\boldsymbol{X}_s$.

### D.6 Visualization of Prediction

We present a prediction showcase on the ETTh2, Electricity, and Traffic datasets, as shown in Figure 8. The predictions closely align with the ground truth, demonstrating that TimeEmb is capable of capturing the complex temporal dependencies of these datasets.

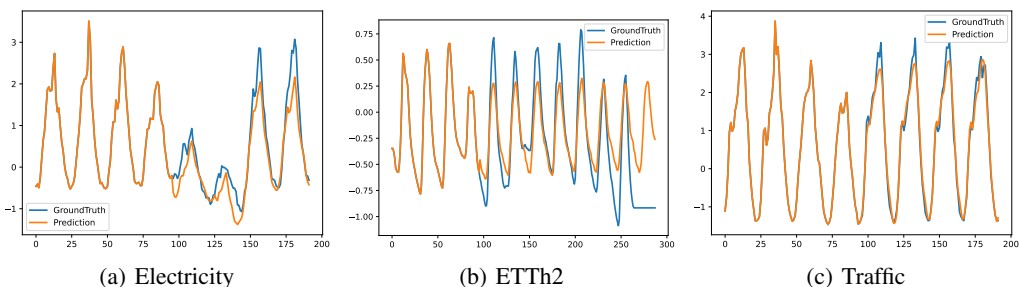

(a) Electricity      (b) ETTh2      (c) Traffic

Figure 8: Visualization of TimeEmb prediction and corresponding groundtruth.

## E  Limitation

While TimeEmb demonstrates strong performance and interpretability in frequency-domain time series modeling, several limitations remain. First, the current design adopts a fixed-resolution embedding bank, which limits its ability to adaptively capture stable patterns across multiple temporal granularities (*e.g.,* hourly, daily, weekly). A more flexible mechanism for multi-scale time-invariant representation would further enhance its capacity for learning multi-periodic structures. Second, the embedding structure is currently discrete, which may restrict its ability to model continuously evolving periodicity. Extending the embedding formulation to a continuous or kernelized representation could enable smoother generalization across unseen temporal slots.

