# OpenReview forum: "TimeEmb: A Lightweight Static-Dynamic Disentanglement Framework for Time Series Forecasting"
_NeurIPS.cc/2025/Conference — NeurIPS 2025 poster_

### Official Review · Reviewer_jvWT · 2025-06-26

**Clarity:** 3
**Significance:** 4
**Originality:** 4
**Rating:** 5
**Confidence:** 5

**Summary:**

The paper introduces an efficient and novel time series forecasting framework TimeEmb. It disentangles time series into static and dynamic components, processed by a learnable embedding bank and a frequency filter, respectively. Extensive experimental results suggest improved performance and efficiency. TimeEmb can also enhance existing methods with trivial extra trainable parameters.

**Questions:**

-	The paper focuses on non-stationarity due to changing trends and external interventions. How would TimeEmb perform on time series characterized by highly irregular or chaotic patterns where traditional periodic assumptions might not hold, or where infrequent, large-magnitude events dominate?

-	Could the authors provide a conceptual diagram or pseudocode illustrating the integration mechanism for TimeEmb when it functions as a ‘plug-in’ module for existing backbone models?

**Ethical Concerns:**

["NO or VERY MINOR ethics concerns only"]

**Final Justification:**

The author's reply addressed all my concerns and I maintained the positive score.

**Limitations:**

Yes

**Quality:**

3

**Strengths And Weaknesses:**

Strengths:

-	TimeEmb pioneers the use of a learnable global embedding bank to capture recurrent and time-invariant patterns across the entire dataset. This approach avoids the rigid assumptions of pre-defined periodicities found in methods like CycleNet.

-	The framework innovatively processes the time-varying component through an efficient frequency-domain filtering mechanism, which is both effective and efficient.

-	The model consistently achieves state-of-the-art or highly competitive performance across seven real-world benchmark datasets and various prediction horizons.

-	TimeEmb exhibits remarkable compatibility, consistently improving the performance of various existing MLP-based and Transformer-based forecasting models when integrated as a plug-in.


Weaknesses:

-	While the embedding bank is claimed to capture "global recurrent features", the description also notes it "provides specific embedding for individual timesteps"  and "time-invariant patterns may differ at different timesteps". This creates a conceptual ambiguity about the precise nature of the "time-invariant" component being learned, blurring the distinction between a truly global pattern and a collection of localized periodic patterns.

-	The paper does not clearly specify how the optimal value of loss weight $\alpha$ is determined for each dataset or scenario.

-	It would enhance the empirical verification to analyze the performance under a distribution-shifting setting.

---

> ### Author Rebuttal · Authors · 2025-07-30
>
> Thanks for your constructive feedback!
> ## W1: The clarification for the conceptual ambiguity about the "time-invariant" component.
> To avoid ambiguity, we define the time-invariant component:
> The "time-invariant" component in TimeEmb refers to periodically recurrent, position-specific(decided by the time index) stability across the dataset, not absolute invariance across all timesteps. The embedding bank achieves "global recurrent features" by learning these stable patterns for each cycle position, ensuring they generalize across all instances of that position in the time series.
>
> ## W2: How the optimal value of loss weight $\alpha$ is determined for each scenario?
> We empirically select the loss weight $\alpha$ from the candidate set {0, 0.25, 0.5, 0.75, 1} and report the optimal results obtained. The detailed values of $\alpha$ are provided in the accompanying code, and a table listing these $\alpha$ values will be included in the main text in subsequent revisions.
>
> ## W3: Analysis of the performance under a distribution-shifting setting.
> As noted in related studies [1], certain datasets—ETTh2, for instance—exhibit marked irregular periodicity and distribution shifts. Our model demonstrates strong performance on such datasets, achieving substantial improvements over existing methods specifically designed to address distribution drift.
>
> ## Q1: How would TimeEmb perform on time series characterized by highly irregular or chaotic patterns?
> As noted in the response to W3, TimeEmb already demonstrates strong performance on datasets with irregular patterns such as ETTh2. Theoretically, even in scenarios of extreme irregularity, TimeEmb can still capture sufficient information: Static components, through learnable embedding banks, capture the underlying long-term stable patterns in data, thereby serving as a reference anchor for chaotic sequences. Even in the absence of traditional periodicity, latent regularities may still be uncovered. In contrast, dynamic components are specifically designed to capture irregular fluctuations and transient changes caused by rare events via frequency-domain filtering, enabling adaptation to abrupt disturbances from non-periodic events.
> Extreme irregular periodicity remains a valuable long-term research direction. In future work, we plan to expand our model to conduct in-depth analysis of this issue and develop targeted solutions.
>
> ## Q2: A conceptual diagram or pseudocode illustrating the integration mechanism for TimeEmb.
> In our model, sequences are processed in both the frequency domain and the time domain. Specifically, operations in the frequency domain can be interpreted as reshaping the sequence spectrum within the original feature space, while in the time domain, a projection layer is employed to generate the final prediction. Building on this framework, the time-domain projection process in our original model serves as a backbone component. When extending our approach to other models, integration is streamlined: only replace our backbone (i.e., the projection layer) with the alternative models. Notably, this modification yields significant performance improvements with minimal additional parameter overhead.
>
> The pseudocode is as follows:
>
> (Choosable) Instance Normalization($X$)
>
> $\overline{X}$ = $rFFT(X)$
>
> $X_d$ = $\overline{X}$ − $X_s$
>
> $\dot{X}$ = $H_ω$ ($X_d$) + $X_s$
>
> $Z$ = $irFFT(\dot{X})$
>
> $Y$ = $Backbone(Z)$
>
> (Choosable) Reverse Instance Normalization($Y$)
>
> where Backbone is referred to the alternative model.
>
> PS: Whether or not to perform Instance Normalization depends on the alternative model.
>
> [1] Shao, Z., Wang, F., Xu, Y., Wei, W., Yu, C., Zhang, Z., ... & Cheng, X. (2024). Exploring progress in multivariate time series forecasting: Comprehensive benchmarking and heterogeneity analysis. IEEE Transactions on Knowledge and Data Engineering.

---

> > ### Comment · Reviewer_jvWT · 2025-08-06
> >
> > Thanks for the authors' informative responses, which address my prior concerns. The TimeEmb framework offers a novel approach to addressing the challenge of non-stationarity by decomposing time series into static (time-invariant) and dynamic (time-variable) components. After reviewing the comments of other reviewers and the author's responses, I agree with its research paradigm and significance.
> > I recognize that TimeEmb's decomposition paradigm aligns partially with the goal of continuous learning. It effectively separates long-term stable patterns from short-term variations. The static embeddings serve as a knowledge base, storing learned long-term patterns and mitigating catastrophic forgetting. The frequency filters in the dynamic components adapt to transient changes, enhancing the model's adaptability in non-stationary environments. This "divide and conquer" strategy enables efficient incremental learning and rapid adaptation to new tasks.
> > The framework's innovation lies not only in its predictive performance but also in its interpretable and scalable new paradigm for dealing with non-stationarity and distributional drift. I believe this research contributes uniquely and deserves acceptance.

---

> > > ### Author Response · Authors · 2025-08-08
> > >
> > > Dear reviewer, thank you very much! We will do our best to incorporate the mentioned new analyses into the main text or the appendix. Once again, we are very grateful for your detailed and constructive review!

---

> ### Author Response · Authors · 2025-08-04
> **A gentle reminder about rebuttal**
>
> Dear Reviewer jvWT:
>
> Thank you once again for your valuable suggestions on our paper – we truly appreciate the insights you’ve shared.
> As we’re now midway through the rebuttal period, we’d be incredibly grateful if you might share your thoughts on whether our response has adequately addressed the concerns you raised. Your feedback would mean a lot to us, as it will help us refine our work further.
> We’re also more than happy to engage in any further discussions and remain ready to clarify any additional questions you may have.
> Thank you again for your time and guidance.
>
> Sincerely,
>
> Paper 20677 Authors

---

### Official Review · Reviewer_5idw · 2025-06-26

**Clarity:** 4
**Significance:** 3
**Originality:** 4
**Rating:** 5
**Confidence:** 4

**Summary:**

This paper proposes TimeEmb, a lightweight framework for time series forecasting by disentangling static and dynamic components. It employs a learnable embedding bank for time-invariant patterns and a frequency filter for time-varying components, primarily operating in the frequency domain. The paper highlights its efficiency and effectiveness on various real-world datasets and its plug-and-play capability.

**Questions:**

Q1. The paper uses ``Instance Normalization’’ at the beginning of the framework. Could the authors elaborate on the impact of this choice compared to other normalization techniques or no normalization, especially in the context of non-stationary time series?

Q2. Please response to the comments in Weaknesses.

**Ethical Concerns:**

["NO or VERY MINOR ethics concerns only"]

**Final Justification:**

Based on the rebuttal discussions, I have adjusted the score accordingly.

**Limitations:**

Yes

**Paper Formatting Concerns:**

There are no formatting concerns in the paper.

**Quality:**

4

**Strengths And Weaknesses:**

Strengths:

S1. This paper proposes a well-motivated disentanglement for time series prediction. The static-dynamic decomposition is a straightforward approach to model long-term patterns separate from short-term fluctuations. Beyond existing research which concatenates learnable time-in-day embeddings, the embedding bank of TimeEmb shares the same embedding space with time series and maintains the common pattern shared across the whole dataset. The well-designed disentanglement representation learning approach yields consistently superior performance beyond diverse state-of-the-art baselines.

S2. The proposed TimeEmb is both effective and efficient. TimeEmb's lightweight architecture is a significant advantage, showcasing competitive performance with fewer parameters and lower training times compared to many strong baselines.

S3. The paper is well-written and generally clear. The figures, especially Figure 1 illustrating performance vs. efficiency, are highly informative and well-designed. The methodology section provides a good overview of the framework and its components. Equations are presented clearly, and the high-level algorithm is provided in Appendix B. The experimental setup and implementation code are detailed, aiding reproducibility.

\
Weaknesses:

W1. Absence of statistical significance reporting for main results: The paper presents mean MSE/MAE values but lacks error bars or statistical significance tests in Tables 1 and 2. While the authors specify averaging over five runs, showing the variance (e.g., standard deviation) would greatly strengthen the empirical claims.

W2. The ablation study shows that removing $H_{\omega}$ leads to notable degradation. However, the paper doesn't delve deeper into how the learnable $\omega$ adapts across different datasets or non-stationary patterns.

W3. There are many related papers on time series forecasting. More through discussion and comparison with these papers are encouraged to be included in the paper.

---

> ### Author Rebuttal · Authors · 2025-07-30
>
> Thanks for your constructive feedback!
> ## W1: Absence of statistical significance reporting for main results.
> The detailed results are as follows.
> |Dataset|ETTh1|ETTh2|ETTm1|ETTm2|Weather|Electricity|Traffic|
> |-|-|-|-|-|-|-|-|
> |Metric|MSE,MAE|MSE,MAE|MSE,MAE|MSE,MAE|MSE,MAE|MSE,MAE|MSE,MAE|
> |96|0.366±0.001,0.387±0.001|0.277±0.001,0.328±0.001|0.304±0.001,0.343±0.001|0.163±0.001,0.242±0.001|0.150±0.001,0.190±0.001|0.136±0.001,0.231±0.001|0.432±0.002,0.279±0.001|
> |192|0.417±0.001,0.416±0.001|0.356±0.001,0.379±0.001|0.354±0.001,0.373±0.001|0.226±0.001,0.285±0.001|0.200±0.001,0.238±0.001|0.153±0.001,0.246±0.001|0.442±0.001,0.289±0.001|
> |336|0.457±0.001,0.436±0.001|0.400±0.002,0.417±0.001|0.379±0.001,0.393±0.001|0.286±0.001,0.324±0.001|0.259±0.001,0.282±0.001|0.170±0.001,0.264±0.001|0.456±0.002,0.295±0.002|
> |720|0.459±0.002,0.460±0.001|0.416±0.001,0.437±0.002|0.435±0.001,0.428±0.001|0.383±0.001,0.381±0.001|0.339±0.001,0.336±0.001|0.208±0.001,0.297±0.001|0.487±0.003,0.311±0.001|
> |Avg|0.425±0.001,0.425±0.001|0.362±0.001,0.390±0.001|0.368±0.001,0.384±0.001|0.265±0.001,0.308±0.001|0.237±0.001,0.262±0.001|0.167±0.001,0.260±0.001|0.454±0.002,0.293±0.001|
>
> ## W2: How the learnable $\omega$ adapts across different datasets or non-stationary patterns?
> Thanks for discussing the valuable point.
> From a design perspective, $\omega$ is a complex-valued vector that selectively weights time-varying frequency bands via element-wise multiplication. This design draws on the convolutional theorem in signal processing, equating frequency-domain filtering to time-domain convolution operations—enabling the model to approximate any linear transformation through end-to-end optimization and adapt flexibly to diverse data.
>
> **Cross-dataset adaptation**:
> - For low-frequency dominant data (e.g., power grids with diurnal patterns), $\omega$ amplifies low-frequency weights to capture stable fluctuations.
> - For high-frequency rich data (e.g., weather with bursts), it prioritizes high-frequency bands to track transient changes.
>
> **Non-stationarity handling**:
> - In stable phases, $\omega$ weakens high-frequency noise for smoothness.
> - During anomalies, it boosts high-frequency weights to capture abrupt shifts.
>
> This adaptive weighting mechanism can be interpreted as learning the frequency response function of a linear time-invariant (LTI) system, enabling the model to robustly handle diverse spectral characteristics and dynamic shifts without the need for manual tuning.
>
> ## W3: More discussion and comparison with time series papers.
> Thanks for your suggestion, we will add two more categories of related discussions—LSTM-based models and LLM-based models to Related Work in
> the next release.
>
> ## Q1: The impact of Instance Normalization in the context of non-stationary time series.
> In time series forecasting, Instance Normalization (IN) differs from Batch Normalization (BN) and Layer Normalization (LN) in key ways. BN normalizes using batch-level statistics, which can blur temporal variations in non-stationary data (e.g., mixing peak and off-peak periods in a batch) . LN normalizes across features for a single sample, potentially smoothing critical local fluctuations. IN, by contrast, normalizes each instance independently at every timestep, preserving instance-specific temporal patterns. This suits non-stationary time series, as it avoids distorting dynamic shifts, providing a stable foundation for subsequent frequency decomposition and embedding learning in TimeEmb.

---

> > ### Comment · Reviewer_5idw · 2025-08-04
> >
> > Thanks for your responses. I have no more questions regarding the paper, and would keep my score.

---

> > > ### Author Response · Authors · 2025-08-04
> > >
> > > Thank you for taking the time to respond to us; your suggestions are of great importance to us.

---

### Official Review · Reviewer_pNtD · 2025-06-30

**Clarity:** 3
**Significance:** 3
**Originality:** 3
**Rating:** 5
**Confidence:** 5

**Summary:**

This paper introduces TimeEmb, a novel and highly effective framework for time series forecasting that addresses temporal non-stationarity by disentangling static (time-invariant) and dynamic (time-varying) components. The core innovation lies in using a learnable global embedding bank for static patterns and an efficient frequency-domain filter for dynamic fluctuations, all primarily operating within the frequency domain. The experimental results are compelling, demonstrating state-of-the-art performance with significantly reduced computational resources, and a remarkable plug-and-play compatibility with existing models. The paper also provides insightful visualizations supporting its disentanglement claims.

**Questions:**

1. The authors employ a fixed-resolution embedding bank to represent time-invariant components. Beyond the current approach of manually adjusting the number of basis vectors ($M$), have the authors explored or considered mechanisms for adaptively learning or inferring optimal multi-scale granularities during training? Such an approach could provide a more holistic solution to the acknowledged limitations related to static representation flexibility.

2. Figures 3 and 4 provide qualitative visualizations to illustrate the separation between static and dynamic components, suggesting improved disentanglement. To further substantiate the interpretability claims, could the authors elaborate on the specific types of insights or actionable information these disentangled components yield in real-world time series applications? For instance, how might practitioners leverage the static vs. dynamic decomposition in forecasting, anomaly detection, or downstream decision-making tasks?

**Ethical Concerns:**

["NO or VERY MINOR ethics concerns only"]

**Limitations:**

See **Weaknesses** and **Questions**.

**Quality:**

4

**Strengths And Weaknesses:**

**Strengths:**

- Proposes a novel disentanglement framework that separates static and dynamic components, addressing limitations in prior time series models.
- Introduces a learnable global embedding bank for time-invariant patterns, improving over traditional local trend-season decompositions.
- Demonstrates strong empirical performance across diverse datasets and baselines, with better efficiency (e.g., significantly fewer parameters than iTransformer).

**Weaknesses:**

- The time embedding bank requires manual tuning of $M$ based on domain knowledge. A fully adaptive, data-driven approach to learning multi-scale structures would improve applicability to real-world time series with nested or irregular periodicities.

---

> ### Author Rebuttal · Authors · 2025-07-30
>
> Thanks for your constructive feedback!
> ## W & Q1: A fully adaptive, data-driven approach to learning multi-scale structures.
> This is an excellent observation that merits further exploration. The embedding bank mechanism proposed in our paper is specifically designed to model regular periodic patterns—such as daily, weekly, or other recurring temporal structures—enabling it to effectively handle tasks characterized by such regular periodicity. Given the constrained search space of regular periodic patterns, manual parameter adjustment (e.g., setting M as discussed earlier) is more efficient in practical applications than fully adaptive learning mechanisms. Moreover, the regular periodic characteristics of most datasets can be pre-analyzed to guide this process. For datasets with irregular periods and mean drift, such as ETTh2 [1], our model still shows excellent performance. However, extreme irregular periodicity remains a valuable long-term research direction, and we plan to expand our model to conduct in-depth analysis of this issue and develop targeted solutions.
>
> ## Q2: Specific types of insights for the disentangled components in real-world time series applications.
> In this paper, we focus on one of the most critical tasks in long-term time series forecasting. Looking ahead, the time-invariant components modeled in our framework hold significant potential for extension to a broader range of related tasks. For instance, they can be leveraged to enhance performance in scenarios such as forecasting with missing values, data imputation, and other temporal data modeling tasks.
> From a practitioner’s perspective, consider the application of our framework in Energy Grid Management:
> Static (time-invariant) component: This component, learned via the embedding bank, captures consistent daily electricity usage patterns—such as recurring morning and evening demand peaks. For practitioners, this insight enables proactive grid resource allocation: for example, activating backup generators or pre-positioning energy reserves ahead of anticipated peak periods to ensure stable supply.
> Dynamic (time-varying) component: By contrast, this component identifies short-term, irregular fluctuations, such as sudden demand spikes triggered by extreme weather events (e.g., heatwaves driving increased air conditioning use). Armed with this real-time information, operators can make agile adjustments to supply—such as ramping up output from renewable energy sources or redistributing power across the grid—to mitigate risks of overloads or blackouts.
>
> [1] Shao, Z., Wang, F., Xu, Y., Wei, W., Yu, C., Zhang, Z., ... & Cheng, X. (2024). Exploring progress in multivariate time series forecasting: Comprehensive benchmarking and heterogeneity analysis. IEEE Transactions on Knowledge and Data Engineering.

---

> ### Author Response · Authors · 2025-08-04
> **A gentle reminder about rebuttal**
>
> Dear Reviewer pNtD:
>
> Thank you once again for your valuable suggestions on our paper – we truly appreciate the insights you’ve shared.
> As we’re now midway through the rebuttal period, we’d be incredibly grateful if you might share your thoughts on whether our response has adequately addressed the concerns you raised. Your feedback would mean a lot to us, as it will help us refine our work further.
> We’re also more than happy to engage in any further discussions and remain ready to clarify any additional questions you may have.
> Thank you again for your time and guidance.
>
> Sincerely,
>
> Paper 20677 Authors

---

> > ### Comment · Reviewer_pNtD · 2025-08-06
> >
> > I appreciate the authors' response, which has addressed my concerns, and I accordingly will maintain my score. Additionally, I suggest including a comparison of training time and parameter counts across datasets, and offering more guidance on selecting the embedding granularity $M$.

---

> > > ### Author Response · Authors · 2025-08-08
> > >
> > > Dear reviewer, thank you very much for your continued feedback and for maintaining your score. We greatly appreciate your valuable suggestions. We will incorporate these improvements in the final paper. Once again, we sincerely thank you for taking the time to review our paper!

---

### Official Review · Reviewer_9Nmb · 2025-07-03

**Clarity:** 2
**Significance:** 3
**Originality:** 2
**Rating:** 4
**Confidence:** 3

**Summary:**

This paper proposes a lightweight static-dynamic decomposition framework for time series forecasting. To obtain optimal performance in the face of distribution shifts, the proposed method learns to separate long-term patterns and short-term fluctuations by separating the time series into two complementary components. The authors demonstrate superior performance and storage efficiency on seven diverse benchmark datasets and functions as an efficient plug-in to enhance existing methods with minimal computational cost.

**Questions:**

Q1) The proposed method captures time-invariant components through an embedding bank and utilizes them to separate time-variant components. However, the method fuses the two separated components just before the model’s output layer.  In contrast, other existing methods, such as Fredformer and DLinear, typically transform each separated feature back into the time domain before performing fusion. Is there a logical or theoretical reason why the proposed method does not adopt a similar design?
Q2) Could the authors clarify how the parameter *M* is determined in each experiment? Is it selected based on prior knowledge, validation performance, or a fixed heuristic?
Q3) While the paper claims that the proposed method can be easily extended to existing models, Table 3 shows only four base models being expanded, and only a subset of datasets used in Table 1 are included. Do the authors have additional experimental results (as seen in appendices for other experiments) demonstrating broader compatibility across more datasets and more base models?
Q4) In Figure 4(b), the time-variant components extracted by the proposed method show clear similarities within the same day of the week and are distinguishable across different days. However, the authors describe the time-variant components as reflecting local fluctuations in the time series, such as anomalies caused by unusual weather or traffic incidents. If such components are indeed day-specific and recurring, wouldn’t it be more appropriate for this structure to emerge in the feature space of the time-invariant components instead? How do the authors reconcile this discrepancy between their definition and the observed clustering pattern?

**Ethical Concerns:**

["NO or VERY MINOR ethics concerns only"]

**Final Justification:**

I have no further concerns. I suggest that the authors provide detailed explanations from the rebuttal in the revised paper for clarity. I'm raising my score to 4.

**Limitations:**

Yes

**Quality:**

2

**Strengths And Weaknesses:**

Strengths
S1) The proposed method achieves competitive accuracy compared to state-of-the-art approaches, despite its simple architecture that does not rely on mechanisms such as attention.
S2) The method is quantitatively evaluated against several recent baselines using benchmark datasets, demonstrating improved performance in time series forecasting. Additionally, ablation studies are conducted to validate the contribution of each component of the proposed method.
S3) The source code is publicly available, allowing for detailed inspection and verification of the implementation.

- Weaknesses
W1) The rationale for fusing the separated time-invariant and time-variant components in the feature space remains unclear. While Section 4.5 emphasizes the importance of high-amplitude or low-frequency components in future prediction, this does not sufficiently justify why the static time-invariant and dynamic time-varying components should be recombined.
W2) The authors criticize an existing method (CycleNet) for assuming a fixed periodic pattern in the dataset, stating that this assumption may not hold under complex or variable periodicities. However, the proposed method similarly relies on a user-defined value of *M* to impose periodicity, and thus suffers from the same limitation—namely, the inability to adapt to varying or complex periodic patterns.
W3) There is a discrepancy between the model described in the paper and the implementation provided in the submitted code. In particular, for the computation described in Section 3.2, the code applies different processing to the real and imaginary parts, yet this distinction is not mentioned in the main text.
W4) The authors claim that their method can be easily extended to enhance existing approaches. However, the paper lacks concrete explanation or examples of how such extensions would be implemented, leaving open the question of whether the method is truly straightforward to apply in practice.

---

> ### Author Rebuttal · Authors · 2025-07-30
>
> Thanks for your constructive feedback!
> ## W1: The rationale for fusing the separated time-invariant and time-variant components in the feature space.
> In our paper, we decompose complex time series into time-varying and time-invariant components, which are then modeled independently to enable each factor to learn a more precise representation. These two factors are recombined to integrate information to derive the final comprehensive forecasting representation.
>
> ## W2: Explanation of TimeEmb's periodicity handling and differences from CycleNet
> First, due to the patterns of human activities, time series data such as electricity consumption and traffic flow have stable periodic basic patterns.
> Second, while existing works have extensively focused on modeling such stable patterns [1], our work diverges by explicitly decoupling time-invariant static patterns (capturing long-term stable periodicities) and time-varying dynamic components (reflecting transient fluctuations). This design choice not only enhances interpretability but also enables more targeted modeling.
> Third, compared to CycleNet, we would like to kindly emphasize that it requires a strong expert knowledge through the RCF algorithm, that is, the length of a stable periodicity. By contrast, we simply model daily patterns, and have achieved optimal results in nearly all experimental settings without the limitation on domain expertise.
> Finally, we fully concur with your emphasis on the importance of modeling complex periodicity in time series analysis. We highlight that the core innovation of this paper lies in proposing an explicitly decoupled framework for complex time series, an efficient, interpretable, and compatible modeling scheme. To address more intricate periodic patterns, we plan to explore adaptive multi-period modeling methods in future work. Thank you again for this insightful feedback.
> For details on the selection of M, please refer to Q2.
>
> ## W3: The clarification for the discrepancy between the model in the paper and the implementation in the code.
> We kindly ask you to specify the inconsistency. In Section 3.2, we transform the original sequence into the frequency domain via rFFT, which is consistent with the implementation provided in the accompanying code.
>
> Regarding the reviewers’ observation on the differential handling of real and imaginary parts, this likely refers to the subtraction of real components described in Section 3.3. To mitigate computational and storage overhead associated with the embedding bank, we leverage the properties of complex conjugation: specifically, we only use the real part of $\overline{X}$ for decoupling. Critically, the real and imaginary components are reintegrated before frequency-domain filtering, aligning with the procedures detailed in this paper(Line 180).
>
> ## W4: The concrete explanation for the extensions of TimeEmb.
> Thanks for pointing it out.
> In our model, sequences are processed in both the frequency domain and the time domain. Specifically, operations in the frequency domain can be interpreted as reshaping the sequence spectrum within the original feature space, while in the time domain, a projection layer is employed to generate the final prediction. Building on this framework, the time-domain projection process in our original model serves as a backbone component. When extending our approach to other models, integration is streamlined: only replace our backbone (i.e., the projection layer) with the alternative models. Notably, this modification yields significant performance improvements with minimal additional parameter overhead. Thanks again, and we will add this introduction in the next version.
>
> ## Q1: The reason for performing fusion before transforming back into the time domain.
> In contrast to existing methods such as DLinear, where each separated feature is projected into the prediction dimension before fusion, our approach prioritizes feature fusion. This design ensures that decoupling and fusion operations are performed within the original feature space, thereby mitigating the risk of information loss that may arise from premature dimensionality changes.
> Thank you for this valuable discussion. We have conducted experiments to validate this approach, and the results confirm that the "project-first-then-fuse" strategy leads to significant performance degradation and incurs computational overhead.
> ||TimeEmb|Proj-Then-Fuse|
> |-|-|-|
> |Dataset|MSE,MAE|MSE,MAE|
> |ETTh1|**0.425,0.425**|0.442,0.433|
> |ETTh2|**0.362,0.390**|0.366,0.393|
> |ETTm1|**0.368,0.384**|0.374,0.388|
> |ETTm2|**0.265,0.308**|0.269,0.312|
> |Weather|**0.237,0.262**|0.245,0.270|
> |ECL|**0.167,0.260**|0.171,0.264|
> |Traffic|**0.454,0.293**|0.476,0.316|
>
> ## Q2: The choice of the embedding bank size M.
> Thanks for highlighting this. We use the embedding bank to learn global time-invariant dataset information. The choice of M is based on common sense and requires no additional expert knowledge. Setting $M = 24$ captures daily patterns; $M = 7$ or $12$ gives weekly or half-daily patterns if needed. The setup of $M$ is straightforward, robust, and knowledge-free.
> |Dataset|ETTh1|ETTh2|ETTm1|ETTm2|Weather|Electricity|Traffic|
> |-|-|-|-|-|-|-|-|
> |M|24(daily)|24(daily)|24(daily)|24(daily)|24(daily)|24(daily) + 7(weekly)|24(daily) + 7(weekly)|
>
> ## Q3: Additional experimental results for the compatibility analysis.
> To verify compatibility, we select representative baselines within our study. As per your requirement, we conducted further experiments on the ETTh2 dataset. These consistently demonstrate superior performance when models are equipped with our TimeEmb. Furthermore, the additional trainable parameters introduced by integrating TimeEmb are negligible.
> |Horizon|96|192|336|720|
> |-|-|-|-|-|
> |Metric|MSE,MAE|MSE,MAE|MSE,MAE|MSE,MAE|
> |Informer|1.642,1.063|5.115,1.847|4.560,1.749|3.490,1.592|
> |+TimeEmb|**1.292,0.905**|**2.803,1.288**|**2.344,1.204**|**2.381,1.248**|
> |Transformer|2.484,1.312|7.652,2.419|5.177,1.941|3.748,1.669|
> |+TimeEmb|**2.254,1.218**|**5.147,1.801**|**3.504,1.555**|**2.726,1.370**|
> |Fredformer|0.293,0.342|0.371,0.389|0.382,0.409|0.415,0.434|
> |+TimeEmb|**0.289,0.335**|**0.358,0.380**|**0.360,0.394**|**0.387,0.421**|
> |CycleNet|0.285,0.335|0.373,0.391|0.421,0.433|0.453,0.458|
> |+TimeEmb|**0.277,0.327**|**0.351,0.375**|**0.399,0.415**|**0.415,0.436**|
> |DLinear|0.339,0.392|0.457,0.463|0.576,0.530|0.773,0.632|
> |+TimeEmb|**0.336,0.388**|**0.419,0.431**|**0.467,0.466**|**0.721,0.607**|
> |iTransformer|0.304,0.353|0.380,0.399|0.420,0.431|0.422,0.443|
> |+TimeEmb|**0.302,0.348**|**0.380,0.397**|**0.419,0.430**|**0.420,0.441**|
> |**Extra Params**|**10729**|**10729**|**10729**|**10729**|
>
> ## Q4: The explanation for the discrepancy between our definition and the observed clustering pattern.
> The observed clustering pattern of time-variant components(${X_d}$) in Figure 4(b) and their day-of-week specificity can be reconciled with our definition by clarifying the granularity and nature of the disentanglement:
>
> First, the time-invariant component(${X_s}$) captured by our embedding bank is designed to model long-term stable patterns across dataset (e.g., general daily rush hours in traffic or hourly electricity consumption baselines). In Figure 4, we explicitly configure the embedding bank with M=7 to capture week-level invariant patterns (i.e., shared characteristics across the same day of the week, such as typical Monday vs. Sunday behaviors). Thus, the core week-level stability is already encoded in ${X_s}$.
>
> Second, the time-variant component reflects residual fluctuations that deviate from these stable patterns. While $X_d$ exhibits day-of-week clustering, this does not imply it is "stable" in the long term. Instead, these clusters represent subtle, context-dependent variations within each day type (e.g., additional electricity consumption on Monday due to factory start-up, and fluctuations in electricity consumption on Sunday due to increased household activities). These variations are not universal enough to be considered part of the invariant $X_s$, as they lack the persistent, dataset-wide consistency required for the time-invariant component.
>
> In essence, the day-of-week clustering in $X_d$ arises from the fact that $X_s$ has already subtracted the most dominant week-level invariants, leaving behind structured but non-stationary residuals that retain contextual dependencies. This aligns with our definition: $X_d$ captures local dynamics that, while partially structured, are not stable enough to be included in the time-invariant component.
>
> [1] Zeng, A., et al. (2023, June). Are transformers effective for time series forecasting?. AAAI, 2023.

---

> > ### Comment · Reviewer_9Nmb · 2025-08-09
> >
> > Thanks for the detailed response. My concerns have been addressed. I will increase my scores.

---

> ### Author Response · Authors · 2025-08-04
> **A gentle reminder about rebuttal**
>
> Dear Reviewer 9Nmb:
>
> Thank you once again for your valuable suggestions on our paper – we truly appreciate the insights you’ve shared.
> As we’re now midway through the rebuttal period, we’d be incredibly grateful if you might share your thoughts on whether our response has adequately addressed the concerns you raised. Your feedback would mean a lot to us, as it will help us refine our work further.
> We’re also more than happy to engage in any further discussions and remain ready to clarify any additional questions you may have.
> Thank you again for your time and guidance.
>
> Sincerely,
>
> Paper 20677 Authors

---

### Decision · Program_Chairs · 2025-09-17

**Decision:**

Accept (poster)

**Comment:**

(a) Summarize the scientific claims and findings of the paper based on your own reading and characterizations from the reviewers
This paper proposes TimeEmb, a lightweight framework for time series forecasting that explicitly disentangles static (time-invariant) and dynamic (time-varying) components. The static part is captured via a learnable global embedding bank, while the dynamic part is modeled through an efficient frequency-domain filter inspired by full-spectrum analysis. The model is lightweight, easily integrable into existing forecasting methods, and empirically outperforms strong baselines on multiple benchmark datasets while requiring fewer parameters and computational resources. Extensive ablation studies and qualitative visualizations are provided to validate the disentanglement and interpretability.

(b) Strengths

Novelty & Motivation: Introduces a clear static-dynamic decomposition framework for time series, improving interpretability and robustness under distribution shifts.

Efficiency: Achieves strong results with significantly fewer parameters compared to transformer-based baselines.

Empirical validation: Demonstrates consistent improvements across diverse datasets, with comprehensive ablations and qualitative analyses.

Reproducibility: Publicly available code supports transparency.

Practicality: Plug-and-play capability shows its utility for enhancing existing models with minimal modifications.

(c) Weaknesses / missing elements

Fusion rationale: Some reviewers questioned the theoretical justification for fusing invariant and variant components directly in the feature space rather than following more standard designs.

Periodicity assumption: While positioned as an advance over CycleNet, the method still requires user-specified embedding granularity (M), which may limit adaptability to irregular or nested periodicities.

Paper–code mismatch: One reviewer flagged an inconsistency regarding handling of real vs. imaginary parts in the frequency domain, though authors clarified during rebuttal.

Extension clarity: The claim that TimeEmb can be broadly applied as a plug-in lacked concrete, step-by-step examples in the initial draft (later clarified in rebuttal).
Overall, these weaknesses are incremental rather than fundamental; they do not undermine the technical soundness.

(d) Reasons for final decision
The reviewers converged toward a positive view after rebuttal. The paper is technically solid, empirically validated, and well-motivated. While not a conceptual breakthrough at the level of an oral/spotlight, the method is practical, interpretable, and efficient, which makes it a valuable contribution to the NeurIPS community. The main reasons for acceptance as a poster are:

Clear methodological novelty in disentangling static/dynamic factors.

Strong empirical gains with lightweight design.

Solid rebuttal that resolved initial concerns.
The decision not to elevate to oral/spotlight is because the contributions, while solid and useful, are evolutionary rather than revolutionary: the framework does not introduce fundamentally new theory or modeling paradigms, and its novelty mainly lies in a careful design choice.

(e) Reviewer discussion and rebuttal

R9Nmb raised concerns about fusion design, periodicity assumptions, model–code consistency, and unclear plug-in extensions. The authors responded with clarifications (e.g., fusion before projection reduces information loss, M selection is simple and knowledge-free, code handling is consistent, and integration with baselines is straightforward). Additional experiments were provided to validate compatibility. After rebuttal, R9Nmb acknowledged concerns were addressed and raised the score.

pNtD found the paper strong but noted manual tuning of embedding size and suggested adaptive mechanisms. The rebuttal clarified that manual settings are pragmatic for common datasets, with adaptivity as future work. Authors also provided concrete use cases for disentangled insights (e.g., energy grid management). The reviewer maintained a positive score.

5idw emphasized the well-motivated disentanglement design, efficiency, and superior performance across baselines. Concerns were relatively minor, and the reviewer was consistently positive.

Overall, the rebuttal period was constructive: authors engaged thoroughly, reviewers acknowledged clarifications, and the final consensus shifted toward acceptance.